Manuscript prepared for Atmos. Chem. Phys.
with version 2014/09/16 7.15 Copernicus papers of the LATEX class copernicus.cls.
Date: 21 August 2019

# Stratospheric ozone trends for 1985–2018: sensitivity to recent large variability

William T. Ball[1,2], Justin Alsing[3,4], Johannes Staehelin[1], Sean M. Davis[5], Lucien Froidevaux[6], and Thomas Peter[1]

[1]Institute for Atmospheric and Climate Science, Swiss Federal Institute of Technology Zurich, Universitaetstrasse 16, CHN, CH-8092 Zurich, Switzerland
[2]Physikalisch-Meteorologisches Observatorium Davos World Radiation Centre, Dorfstrasse 33, 7260 Davos Dorf, Switzerland
[3]Oskar Klein Centre for Cosmoparticle Physics, Stockholm University, Stockholm SE-106 91, Sweden
[4]Physics Department, Blackett Laboratory, Imperial College London, SW7 2AZ, UK
[5]NOAA Earth System Research Laboratory Chemical Sciences Division, Boulder, CO, USA
[6]Jet Propulsion Laboratory, California Institute of Technology, Pasadena, CA, USA

*Correspondence to:* W. T. Ball (william.ball@env.ethz.ch)

**Abstract.** The Montreal Protocol, and its subsequent amendments, has successfully prevented catastrophic losses of stratospheric ozone, and signs of recovery are now evident. Nevertheless, recent work has suggested that ozone in the lower stratosphere (<24 km) continued to decline over 1998–2016, offsetting recovery at higher altitudes and preventing a statistically significant increase in quasi-global (60°S – 60°N) total column ozone. In 2017, a large lower stratospheric ozone resurgence over less than 12 months was estimated (using a chemistry-transport model; CTM) to have offset the long-term decline in the quasi-global integrated lower stratospheric ozone column. Here, we extend the analysis of space-based ozone observations to December 2018 using the BASIC$_{SG}$ ozone composite. We find that the observed 2017 resurgence was only around half that modelled by the CTM, was of comparable magnitude to other strong inter-annual changes in the past, and restricted to southern hemisphere mid-latitudes (SH; 60°S–30°S). In the SH mid-latitude lower stratosphere, the data suggest that by the end of 2018 ozone is still likely lower than in 1998 (probability ~80%). In contrast, tropical and northern hemisphere (NH) ozone continue to display ongoing decreases, exceeding 90% probability. Robust tropical (>95%, 30°S–30°N) decreases dominate the quasi-global integrated decrease (99% probability); the integrated tropical stratospheric column (1–100 hPa, 30°S–30°N) displays a significant overall ozone decrease, with 95% probability. These decreases do not reveal an inefficacy of the Montreal Protocol. Rather, they suggest other effects to be at work, mainly dynamical variability on long or short timescales, countering the positive effects of the Montreal Protocol on stratospheric ozone recovery. We demonstrate that large inter-annual mid-latitude (30°–60°) variations, such as the 2017 resurgence, are driven by non-linear quasi-biennial oscillation (QBO) phase-dependent seasonal variability. However, this variability is not represented in current regression analyses. To understand if observed lower stratospheric ozone decreases are a transient or long-term phenomenon, progress needs to be made in accounting for this dynamically-driven variability.

## 1 Introduction

Ozone in the stratosphere acts as a protective shield against ultraviolet radiation that may harm the biosphere, and leads to cataracts, skin damage, and skin cancer in humans (Slaper et al., 1996; WMO, 2014, 2018). In the latter half of the 20[th] century, the emission of long-lived halogen-containing ozone depleting substances (hODSs) led to ~5% loss in quasi-global (60°S–60°N) integrated total column ozone (WMO, 2014), which represents the combined changes in tropospheric and stratospheric ozone contributions. The 1987 Montreal Protocol and its amendments and adjustments led to a reduction in hODSs that resulted in a halt in total column ozone losses around 1998–2000 (Harris et al., 2015; Chipperfield et al., 2017).

However, there is still no evidence of a statistically significant increase in total column ozone since 1998 (Chipperfield et al., 2017; Weber et al., 2018; Ball et al., 2018), despite a significant

increase in upper stratospheric ozone (1–10 hPa) (Ball et al., 2017; Steinbrecht et al., 2017; Ball et al., 2018; Petropavlovskikh et al., 2019). Ball et al. (2018) and Ziemke et al. (2018) presented evidence, using OMI/MLS tropospheric column observations for 2005–2016, that tropospheric ozone had also increased significantly. However, large uncertainties remain in quasi-global tropospheric ozone trends, and the recent Tropospheric Ozone Assessment Report (TOAR) shows that different tropospheric ozone products give a wide range of trends, some even indicating negative changes (Gaudel et al., 2018). The importance of considering tropospheric and stratospheric changes separately to understand changes in total column ozone has also been highlighted in recent studies using chemistry climate models (CCMs) (Meul et al., 2016; Keeble et al., 2017; Dhomse et al., 2018). If tropospheric and upper stratospheric ozone have indeed both increased, then the observed flat trend in total column ozone implies that middle and lower stratospheric ozone should have decreased.

To assess trends in stratospheric ozone, composites of observations must be formed by merging multiple ozone observational timeseries into a long, multi-decadal record from which variability can be attributed, and long-term trends determined. Composites are subject to artefacts from merging different observing platforms. Multiple papers (Tummon et al., 2015; Harris et al., 2015; Steinbrecht et al., 2017; Ball et al., 2017, 2018) and a SPARC report (Petropavlovskikh et al., 2019) review, discuss, and attempt to account for the artefacts in the uncertainty budget.

Ball et al. (2018) integrated ozone over the whole stratosphere, i.e. the ozone layer, quasi-globally for pressure levels from 147–1 hPa (~13–48 km) at mid-latitudes (30°–60°), and 100–1 hPa (~16–48 km) between the sub-tropics (30°S–30°N), and found ozone to be lower in 2016 than in 1998 in multiple ozone composites. In their analysis, the lower stratosphere (147/100–32 hPa, ~13/17–24 km) was driving this decrease. The most significant decreases were in the tropics, but negative trends extended out into the mid-latitudes (Fig. 1d). Other studies have subsequently confirmed these negative trends (Zerefos et al., 2018; Wargan et al., 2018; Chipperfield et al., 2018). Evidence points towards dynamical variations driving changes (Chipperfield et al., 2018), perhaps in the form of enhanced isentropic mixing (Wargan et al., 2018).

Part of the negative trends in northern hemispheric stratospheric ozone in the 1980s and 1990s at higher latitudes have been previously attributed to synoptic and planetary waves (Hood and Zaff, 1995; Hood et al., 1999) inducing large localised (e.g. over Europe) wintertime decreases in ozone. These changes in wave activity might be driven by sea surface temperature and eddy flux changes on decadal or longer timescales, although most of these studies are limited to the end of the last century when ODSs remained an established primary driver of the decrease. Since recent studies almost exclusively consider zonal mean ozone fields, this motivates re-investigation of longitudinal ozone changes (in future work). The El Niño Southern Oscillation (ENSO) and Quasi-Biennial Oscillation (QBO) are known to influence the dynamical variability in the lower stratosphere and may be a main player in driving inter-annual and decadal variability in this region (Diallo et al., 2018, 2019). Nevertheless, these dynamical changes do not in themselves determine a specific underlying driving force,

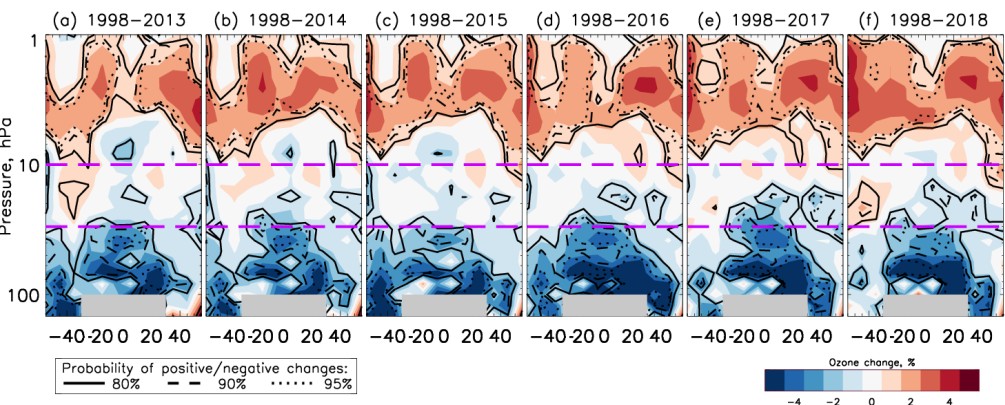

Figure 1: Zonally averaged ozone changes between 1998 and end years (a) 2013 to (f) 2018. Red represents increases, blue decreases (%; right legend). Contours represent probability levels of positive or negative changes (left legend). Grey-shaded regions represent unavailable data. Pink dashed-lines delimit regions integrated to partial ozone columns in other figures.

however the effect of increasing anthropogenic greenhouse gases (GHGs) (Hood and Soukharev, 2005; Peters and Entzian, 1999) on specific mechanisms needs further study (Ball et al., 2018). On the other hand, Stone et al. (2018) showed that negative ozone trends could be simulated in the lower stratosphere over the same period in two of nine ensemble members of a coupled CCM as a result of natural variability interfering in the (linear) trend analysis, although none of the ensembles displayed the same widespread negative trends as detected in observations (Ball et al., 2018). They suggested that an additional seven years of observations would lead to negative signals disappearing in favour of positive trends. The implication, then, is that the observed negative trend over the relatively short 19 year timeframe may be a temporary result from large natural variability (in the single realisation) of the real-world, rather than a response to increasing GHGs.

Chipperfield et al. (2018) used a chemistry transport model (CTM) to reconstruct ozone variability close to past real-world behaviour; transport in the CTM is driven by ERA-Interim (Dee et al., 2011) reanalysis fields. The results showed changes similar to those presented by Ball et al. (2018) up to December 2016. They extended their CTM analysis by an additional 12 months to find that the 1998–2016 ozone decline in the lower stratosphere (~2 DU; Ball et al. (2018)) was offset by a sudden increase of ozone in 2017, exceeding 8 DU quasi-globally. This was attributed almost entirely to dynamical changes and was primarily located in the southern hemisphere (SH). Froidevaux et al. (2019) have also noted that ozone trends derived from Aura/MLS data over a shorter period (2005–2018) have a tendency towards slightly positive values in the SH, but not so elsewhere within the extra-polar regions. Chipperfield et al. (2018) suggested that the lower stratospheric ozone decrease was a result of large natural variability that biased the trend analysis, and that the variability could be attributed to dynamics and not to chemical or photolytic changes, although the source of dynamical

perturbations was not identified or the impact on trends quantified. Thus, an assessment of this recent variability on trends, and an update to 2018 is needed and is a key aim of this study.

Here, we update the observational analysis of Ball et al. (2018) to include data to the end of 2018 (Section 3.1). This allows us to assess the impact of the 2017 ozone increase in the lower stratosphere on the trend analysis, and to consider additional changes over 2018. We show that

large ozone-increase events, with a duration and magnitude similar to that of 2017 (Chipperfield et al., 2018) have occurred regularly since 1985 at mid-latitudes (Section 3.2), and find the events are linked to a seasonally-dependent QBO effect (Section 3.3). We update partial column ozone trends from 2016 to 2018 in section 3.4. Finally, we consider the sensitivity of trends to the recent increase of stratospheric ozone (Sections 3.5-3.6) by considering six periods that start in 1985 and

end between 2013 and 2018, in order to demonstrate where signals are robust to the end date, and where not. Such an analysis is essential to establish if the negative trends are a result of natural variability interfering in the trend analysis, and to take the first steps to account for what might be driving the large, short-term variability.

## 2    Data and methods

### 2.1    Ozone Data

Although other ozone composites exist (Petropavlovskikh et al., 2019), we focus exclusively on data formed by merging ozone from SWOOSH (Davis et al., 2016) and GOZCARDS (Froidevaux et al. (2015); here we use the v2.20 update of Froidevaux et al. (2019)) using the so-called BASIC (BAyeSian Integrated and Consolidated) approach (Ball et al., 2017) to account for arte-

115 facts in merged composites and improve trend estimates. These data were referred to as 'Merged-SWOOSH/GOZCARDS' by Ball et al. (2018), but we refer here to it as $BASIC_{SG}$. To briefly place the SWOOSH and GOZCARDS datasets in the context of $BASIC_{SG}$, Figure S1 of Ball et al. (2018) presented 1998–2016 changes using SWOOSH or GOZCARDS alone; this figure reveals that these ozone composites show generally similar changes on large spatial scales, though there are clear dif-

120 ferences on small scales, e.g. in the tropical upper stratosphere, and in the SH lower stratosphere. Figure S2 of Ball et al. (2018) importantly demonstrates at 100 hPa in the tropical lower stratosphere that there are significant differences between SWOOSH and GOZCARDS in the late 1990s; this figure also shows that $BASIC_{SG}$ is able to account for the differences in a principled way that is not simply the averaging of the two products, which is particularly important for having confidence in

an assessment of ozone in the lower stratosphere. We extend $BASIC_{SG}$ from Ball et al. (2018) by two years to cover 1985–2018. This period is essentially an extension of the Aura Microwave Limb Sounder (Aura/MLS); both SWOOSH and GOZCARDS consider Aura/MLS exclusively after 2009.

We only consider $BASIC_{SG}$ here for the following reasons. First, as discussed in Ball et al. (2018), compared to the other composites it had the least apparent artefacts within the timeseries. The

130 Stratosphere-troposphere Processes And their Role in Climate (SPARC) Long-term Ozone Trends and Uncertainties in the Stratosphere (LOTUS) report (Petropavlovskikh et al., 2019) indicates this method to be more robust to outliers than other composites. Second, BASIC$_{SG}$ is resolved in the lower stratosphere, which is not the case for all composites. For further discussion see Ball et al. (2018) and the SPARC LOTUS report (Petropavlovskikh et al., 2019). Additionally, SWOOSH

and GOZCARDS are currently two of the most up-to-date composites available. Finally, we are interested here in the sensitivity of stratospheric ozone changes to different end years and, since Aura/MLS is arguably one of the best remote sensing platforms for ozone currently in operation (Petropavlovskikh et al., 2019), focusing only on BASIC$_{SG}$ provides an analysis, discussion, and interpretation that is free from the complications of considering multiple composites that have mul-

tivariate reasons for displaying different behaviour.

## 2.2   Regression analysis

As in Ball et al. (2018), we perform all timeseries analysis using dynamical linear modelling (DLM) (Laine et al., 2014) using the public DLM code DLMMC (available at https://github.com/justinalsing/dlmmc). We provide a short overview of DLM here.

Our DLM approach models the ozone timeseries as a (dynamical) linear combination of the following components. There are two seasonal components (with 6- and 12-month periods respectively), a set of regressor variables (i.e., proxy timeseries describing various known drivers), an auto-regressive (AR) process, and a smooth non-linear (non-parametric) background trend. DLM differs from traditional multiple linear regression (MLR) approaches in a number of key ways.

Firstly, while MLR fits for a fixed (constant-in-time) linear combination of seasonal, regressor, and trend components, DLM can allow the amplitudes of the various components to vary dynamically in time, capturing richer phenomenology in the data. Here, we allow the amplitude and phase of the seasonal components to be dynamic, but keep the regressor amplitudes constant in time. We do this because the seasonal cycle in the observational composites can change over time, either as a physical

feedback of changing temperature and ozone or due to different observations exhibiting different seasonal amplitudes (not shown) that are a result of the observing instruments 'seeing' slightly different parts of the atmosphere or having different sampling. Due to the seasonal cycle having the largest variability of all modes we expect that, if left unaccounted for, the time varying seasonal modulation might have an influence on the regression. In principle other regressor amplitudes could also have

some time modulation for similar reasons. However, we leave an investigation of more flexible DLM models with dynamic regressor amplitudes to future work where a physically-motivated justification for such freedom can be investigated.

Secondly, MLR that does not assume a driver for the long-term trends, e.g. for the influence of ODSs or GHGs, typically assumes a fixed prescription for the shape of the background trend, e.g.

a piecewise-linear or independent-linear trend with some fixed, pre-chosen inflection-date. These

assumptions are both restrictive and give a poor representation of the smooth background trends we expect from nature (Laine et al., 2014; Ball et al., 2017). DLM addresses this by instead modelling the trend as a smooth, non-parametric, non-linear curve, where the 'smoothness' of the trend is controlled by a free parameter that is included in the fit (see supplementary materials Fig. S1).

Thirdly, in practice MLR is often performed by first subtracting an estimated mean seasonal cycle, fitting the trend and regressor variables to the anomalies, and then making a post-hoc correction for AR residuals, although many do fit annual and semi-annual components. This procedure typically does not propagate the errors on the seasonal cycle and AR parameters in a rigorous way, leading to misrepresentation of uncertainties. DLM addresses this by inferring all components of the model

simultaneously, and formally marginalizing over the uncertainties in all other parameters when re-porting uncertainties on, e.g., the trend. We use the same prior assumptions as described in Ball et al. (2018).

        Probabilities of an overall increase (decrease) in ozone between two dates (Figs. 1, 7, and Table 1) are computed as the fraction of Monte Carlo Markov Chain (MCMC) samples that show positive

(negative) change. Credible intervals (Figs. 6, 8, 9) are computed as the central 95 and 99 percentiles of the MCMC samples. The use of 'confidence' or 'significance' is used in this paper interchangeably with 'probability' and refers specifically to Bayesian probabilities; it does not refer to the application of frequentist significance tests and/or confidence intervals.

        We use the same regressors as Ball et al. (2018): solar (30 cm radio flux, F30) (Dudok de Wit

et al., 2014)), volcanic (latitudinally resolved stratospheric aerosol optical depth, SAOD) (Thomason et al., 2017), ENSO (NCAR, 2013), and the Quasi-Biennial Oscillation, QBO, at 30 and 50 hPa [1]. In previous analyses, we considered the Arctic and Antarctic Oscillation, AO/AAO [2], as proxies for northern hemisphere (NH) and SH surface pressure variability only for partial column ozone analysis in their respective hemisphere; here we also consider them for the spatially-resolved analysis and in

all cases use both AO and AAO simultaneously. The AO and AAO have little affect outside their respective regions, but we do not limit the possibility they may influence some variability in either hemisphere (Tachibana et al., 2018). We use a first order AR (AR1) process (Tiao et al., 1990) to consider auto-correlation in the residuals. We remove a three year period following the Pinatubo eruption, i.e. June 1991 to May 1994, which is a year longer than the previous analysis, to avoid

any effects of the eruptions that may have persisted. Another key point regarding the SAOD proxy is that, unlike the other proxies that have been fully updated to the end of 2018 for this analysis, the SAOD is currently not extended beyond 2016, so we repeat the year 2016 for 2017 and 2018. If any deviations in the SAOD occurred during the 2017–2018 period our analysis will not account for this. Nevertheless, as can be seen in Fig. 1d here, in comparison to Fig. 1b of Ball et al. (2018), all of

---

[1]QBO indicies: http://www.geo.fu-berlin.de/met/ag/strat/produkte/qbo/

[2]AO/AAO indicies: http://www.cpc.ncep.noaa.gov/products/precip/CWlink/

these adjustments to the procedure from Ball et al. (2018) have little impact on the estimated mean changes in ozone.

## 3  Results

### 3.1  Stratospheric ozone changes since 1998

Figure 1d shows the pressure-latitude, spatially-resolved 1998–2016 ozone change, reproducing
Fig. 1b of Ball et al. (2018). Minor differences exist because the BASIC$_{SG}$ composite and DLM procedure have been updated. Ozone in the lower stratosphere (delimited by the pink dashed line at 32 hPa, or 24 km) shows a marked and almost hemisphere-symmetric decrease, while upper stratospheric changes (>10 hPa, 32 km) are mainly positive. The middle stratosphere generally shows relatively flat ozone trends since 1998 with low probability of an overall change.

Figures 1e and f show the 1998–2017 and –2018 ozone changes, respectively. Four points of interest emerge from a comparison to 1998–2016: (i) while still negative, the magnitude of the lower stratospheric SH (60°S–30°S) ozone decrease has become smaller and less significant; (ii) tropical (30°S–30°N) and NH (30°N–50°N) changes remain negative and highly probable; (iii) the probability (and magnitude) of negative ozone trends over tropical and NH regions in the middle stratosphere
(32–10 hPa) has increased; and (iv) the magnitude and probability of upper stratospheric ozone increase has strengthened. Importantly, Fig. 1 demonstrates the robustness of negative ozone trends in the lower, and positive trends in the upper, stratosphere irrespective of the final year of the analysis. Figures 1a–f present ozone changes from 1998 to end years 2013 through 2018, showing the sensitivity of ozone trends to six consecutive end years. These end years give insight into the sensitivity
of the trends to large inter-annual variability. In particular, these six years encompass periods of both negative/Easterly and positive/Westerly phases of ENSO/QBO. These modes are major contributors to stratospheric variability (Zerefos et al., 1992; Tweedy et al., 2017; Toihir et al., 2018; Garfinkel et al., 2018; Diallo et al., 2018, 2019), and any sensitivity of the end year to the state of these drivers should be encapsulated in the set of spatial responses depending on the end year only (Fig. 1), par-
ticularly if these modes were not well-captured by DLM predictors. A lower stratosphere negative ozone trend is persistent for all end years. For 1998–2013, there is a highly probable negative trend in ozone in the SH lower stratosphere; the probability is retained until 2016, after which it reduces. The opposite is seen in the NH, where only a small region of probable ozone decrease exists for 1998–2013, and this strengthens with each panel until 2016, after which a highly probable decrease
of ozone remains stable. There is no apparent switch from negative to positive ozone changes in these regions for any of the six end years.

The reduced probability of a SH decrease is related, as we will see in Section 3.2, to the rapid 2017 increase in SH mid-latitude lower stratospheric ozone reported by Chipperfield et al. (2018) using a CTM. However, Fig. 1 also confirms in observations that this is localised to south of 30°S and

does not reveal coherent or consistent behaviour over time with the NH, suggesting that there may be large, hemispherically independent variability interfering with the trend estimates. Nevertheless, there are no signs as yet of an ozone increase underway in the quasi-global lower stratosphere.

Further, the decrease in ozone in the tropical lower stratosphere increases in magnitude and significance as more data are added. The tropical lower stratospheric ozone is projected to decrease by the end of the century in all CCMs (Dhomse et al., 2018), due to enhanced upwelling from the Brewer Dobson circulation (BDC) as a result of changes to stratospheric dynamics from increasing GHGs (Polvani et al., 2018). It is possible that this is a detection in observations of the expected tropical lower stratosphere decline in ozone, earlier than expected (WMO, 2014). However, whilst the data show a significant decline, it remains to be seen if this can be attributed to the anthropogenic GHG induced upwelling of the BDC.

### 3.2 On the rapid increase of ozone in 2017

Chipperfield et al. (2018) reported a rapid increase in the quasi-global lower stratospheric ozone in 2017, modelled using a CTM driven by ERA-Interim reanalysis to represent dynamical variability closer to that which occurred historically. The quasi-global, deseasonalised timeseries from BASIC$_{SG}$ is shown in Fig. 2a. The year 2017 is bounded by the vertical dashed lines and the large increase is highlighted in red from a minimum in November 2016 to a maximum reached 11 months later in October 2017.

The observed 2016–2017 increase in Fig. 2a was 5.5 DU, which is 63% of the 8.7 DU increase reported by Chipperfield et al. (2018). Split into three latitude bands, 60°–30°S, 30°S–30°N, and 30°–60°N (Figs. 2b–d), we find that the rapid increase can be decomposed into a 12 DU increase in the SH, 3 DU in the tropics, and 6 DU in the NH. Weighting for latitude – 21, 58, and 21% respectively – the SH contribution accounts for nearly half of the quasi-global increase (2.5 DU, 1.9 DU, 1.3 DU). The overall increase is composed of two sub-periods, dominated by a NH increase until May 2017, and a SH increase over April–August 2017. The tropical region saw comparatively little change in the second period. We do not know why the CTM and observations disagree in the magnitude of change for this period.

Importantly, the rapid increase seen in 2017 is not unique. Four other quasi-global 'events' of this type are found over 1985–2018, shown in Fig. 2a. The identification criterion for these events was an increase of at least 90% of the 2017 increase occurring within a 13 month period. The decomposed timeseries (Fig. 2b–d) show that the large increases in the SH are *normal*, occurring regularly. They also occur in the NH, but not as regularly, and the tropical variability is much smaller than the mid-latitude variance. In addition to the large increases, there are also comparatively large negative swings in both SH and NH timeseries – one in the NH beginning in 2002 exceeds 24 DU. In the following section we argue that these large, rapid changes are driven by a non-linear seasonal-QBO effect.

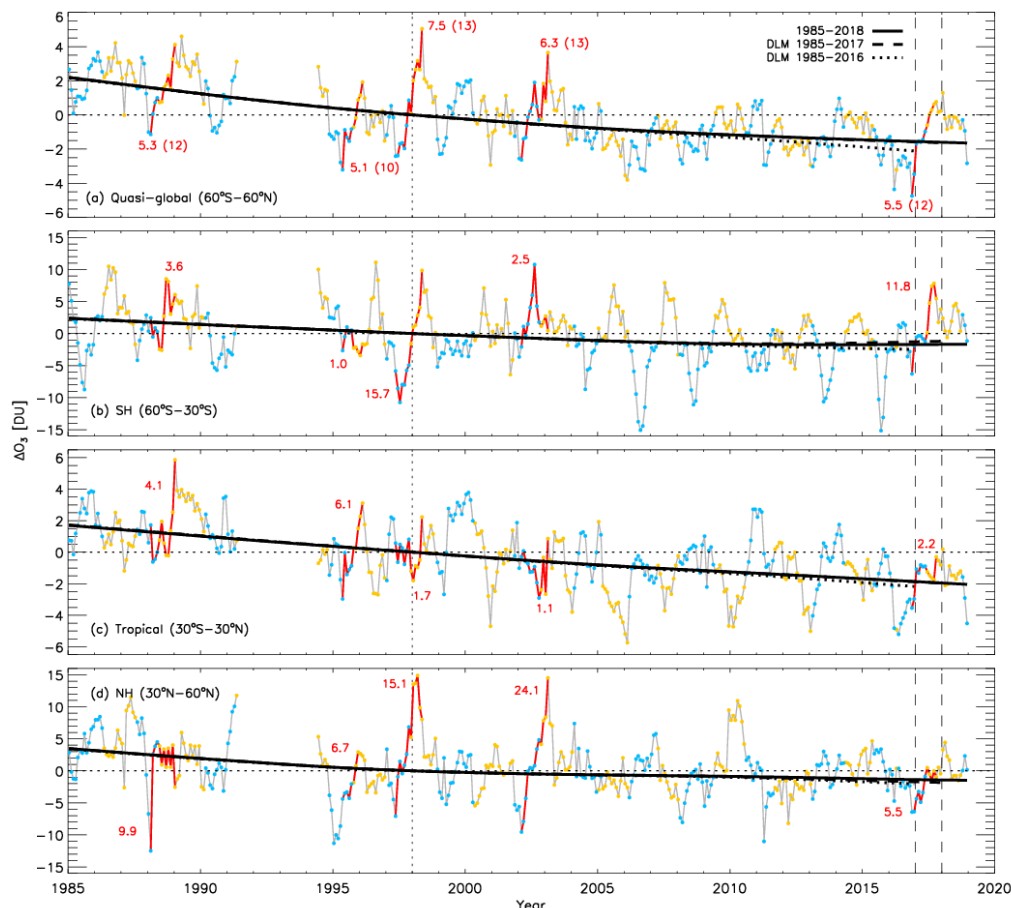

Figure 2: Lower stratospheric partial column ozone anomalies, (a) quasi-global (60°S–60°N), (b) southern hemisphere (60°S–30°S, 147–30 hPa), (c) tropics (30°S–30°N, 100–30 hPa), (d) northern hemisphere (30°N–60°N, 147–30 hPa). The DLM non-linear trend is shown for 1985–2016 (dotted), 1985–2017 (dashed), and 1985–2018 (solid). Red lines represent contiguous periods identified in the quasi-global anomalies exceeding 90% of the magnitude of the November 2016 to October 2017 change within a 13 month period; the DU changes are written above or below each period; red periods in (b–d) are those identified in (a). Colour dots are plotted on each timeseries when the QBO at 30 hPa is either in an Easterly (yellow) or Westerly (blue) phase. The three-year period following the eruption of Mt. Pinatubo, June 1991 to May 1993, has been removed. Figures for the whole, upper and middle stratosphere are provided in the Supplementary Materials Figs. S2–4.

### 3.3 Contribution of QBO to mid-latitude ozone variability

Chipperfield et al. (2018) convincingly showed that the majority of post-1997 quasi-global lower stratospheric ozone variability in Fig. 2a was dynamically controlled, specifically in the lower stratosphere where the lifetime of ozone is long. Given that the contributions from each sub-region

(Fig. 2b–d) add up to the quasi-global change in 2017, it is reasonable to assume that dynamics controls much of the sub-decadal variability there too. The peaks (or troughs) in the SH are 2–3 years apart. The QBO has a similar periodicity and is known to have the largest inter-annual dynamical impact on ozone in the stratosphere (see Gray and Pyle (1989), Zerefos et al. (1992), and Toihir et al. (2018), and references therein). Labelling each month in Fig. 2 with the 30 hPa QBO-Easterly or Westerly phase in yellow or blue dots, respectively, reveals that the large SH negative anomalies are almost always associated with a Westerly phase, while positive anomalies are associated with an Easterly phase; Bodeker et al. (2007) previously identified large SH negative anomalies in 1985, 1997 and 2006 and related these to the QBO-Westerly phase. This also appears to be the case in the NH, but the variability is less regular, unsurprisingly since the NH stratosphere is known to have additional variability as a consequence of greater sea-land contrast and more orography than in the SH. The NH thus exhibits stronger large scale wave activity and consequently polar vortex and stratospheric variability (see Butchart (2014) and Kidston et al. (2015) and references therein). Equatorial variability in ozone related to the QBO phase at 30 hPa shows the opposite behaviour to that at mid-latitudes: decreases in ozone generally appear to occur with the Easterly phase and vice versa, and the return from maximum excursion (i.e. the sign of the gradient) appears to be more related to the change in phase.

Histograms of the ozone anomalies relative to the DLM trend line for each QBO phase at 30 hPa are shown in the upper row of Fig. 3. The shift in the histogram between QBO phases is clear in the SH. The NH displays little shift, again likely related to other drivers influencing NH ozone changes, though the extremes show a similar phase separation as in the SH. The difference between the QBO Easterly and Westerly histograms are shown in the bottom row, and make clear the correlation between the QBO state and ozone anomalies.

To clarify this further, in Fig. 4 all 34 years in the 1985–2018 period are split into 13-month periods starting in January for the SH (upper row) and July for the NH (lower row), i.e. a few months prior to the onset of winter in the respective hemisphere. The latitudes plotted are refined to isolate clearer signals for (a) 50°–30°S, (c) 30°–50°N, and (b,d) 10°S–10°N. This refinement reduces the influence of polar variability on the 30°–50° band, and isolates the equatorial region to where the QBO variability is strongest. We note that the act of forming partial columns of ozone may reduce the integrated variability compared to counter-varying layers that would otherwise be resolved by pressure level. We find the use of the QBO phase at 15 hPa also better separates the events in this additional analysis. We find negative and positive ozone excursions in the lower stratosphere become clear in 13-month segments when they are bias-shifted to zero in March (a,b) and September (c,d) and then colour coded according to their QBO phase in April or October, respectively (vertical dotted line). The largest deviations are found to occur four or five months later (vertical dashed line), at the onset of hemispheric autumn (Holton and Tan, 1980; Dunkerton and Baldwin, 1991). It is also interesting to note that the only large, positive QBO-Westerly anomaly that peaks four

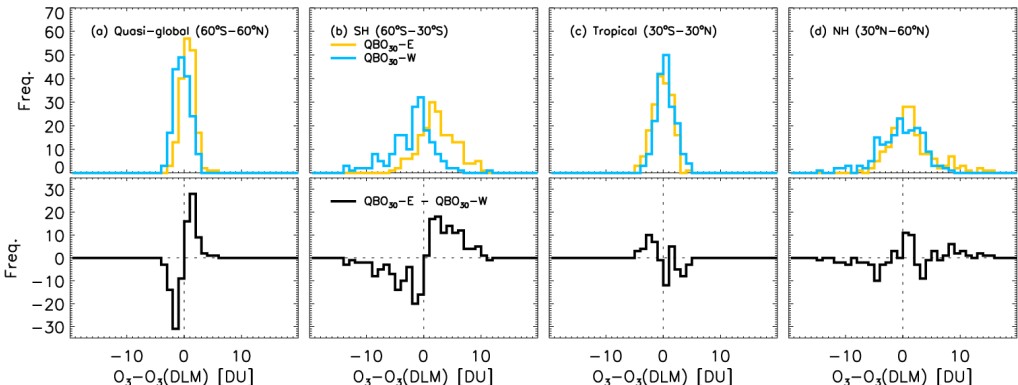

Figure 3: (**Upper row**) Histograms of ozone anomalies relative to the DLM non-linear trend line in Fig. 2 for months when the QBO at 30 hPa is either in an Easterly (yellow) or Westerly (blue) phase: (a) quasi-global (60°S–60°N), (b) southern hemisphere (60°–30°S, 147–30 hPa), (c) tropics (30°S–30°N, 100–30 hPa), (d) northern hemisphere (30°–60°N, 147–30 hPa). (**Lower row**) Difference between QBO Easterly and Westerly histograms from the upper row.

months later, in either hemisphere, occurs in the SH in 2002. This year is famous for having the only observed sudden stratospheric warming in the SH, and indicates that while the QBO-phase appears to dominate this distribution of anomalies, other processes can also sometimes dominate.

315 We reiterate that the separation of positive and negative anomalies into those related to Easterly or Westerly QBO phases is clearest for the SH (Fig. 4a) and the corresponding, opposing, equatorial changes (Fig. 4b). The anti-correlated behaviour of anomalies between mid-latitude and equatorial regions is consistent with previous studies investigating the relationship between the QBO and mid-latitude ozone variability (e.g. Zerefos et al. (1992); Randel et al. (1999); Strahan et al. (2015)). We

320 summarise the dynamical concept, in the context of these results, in the following (see Baldwin et al. (2001) and Choi et al. (2002) for detailed discussion). The QBO consists of downward propagating equatorial zonal winds; in the lower stratosphere this consists of a Westerly above an Easterly, or vice versa. A Westerly above Easterly (i.e. the 15 hPa QBO is Westerly as identified by blue lines in Fig. 4) leads to a shear that induces an anomalous downward motion of air, and adiabatic warming

325 (Fig. 1 of Choi et al. (2002)) and also leads to an anomalous increase in ozone. For an Easterly above a westerly, this leads to anomalously rising air and adiabatic cooling together with an associated ozone decrease. An equator-to-mid-latitude circulation forms to conserve mass (Randel et al., 1999; Polvani et al., 2010; Tweedy et al., 2017). At sub-tropical and mid-latitudes, the return of this meridional circulation draws ozone-rich air from above down into ozone poor regions, anomalously

330 enhancing ozone there (yellow, Fig. 4a,c). When Easterlies lie over Westerlies (blue, Fig. 4), the opposite circulation is set up, and ozone anomalies reverse.

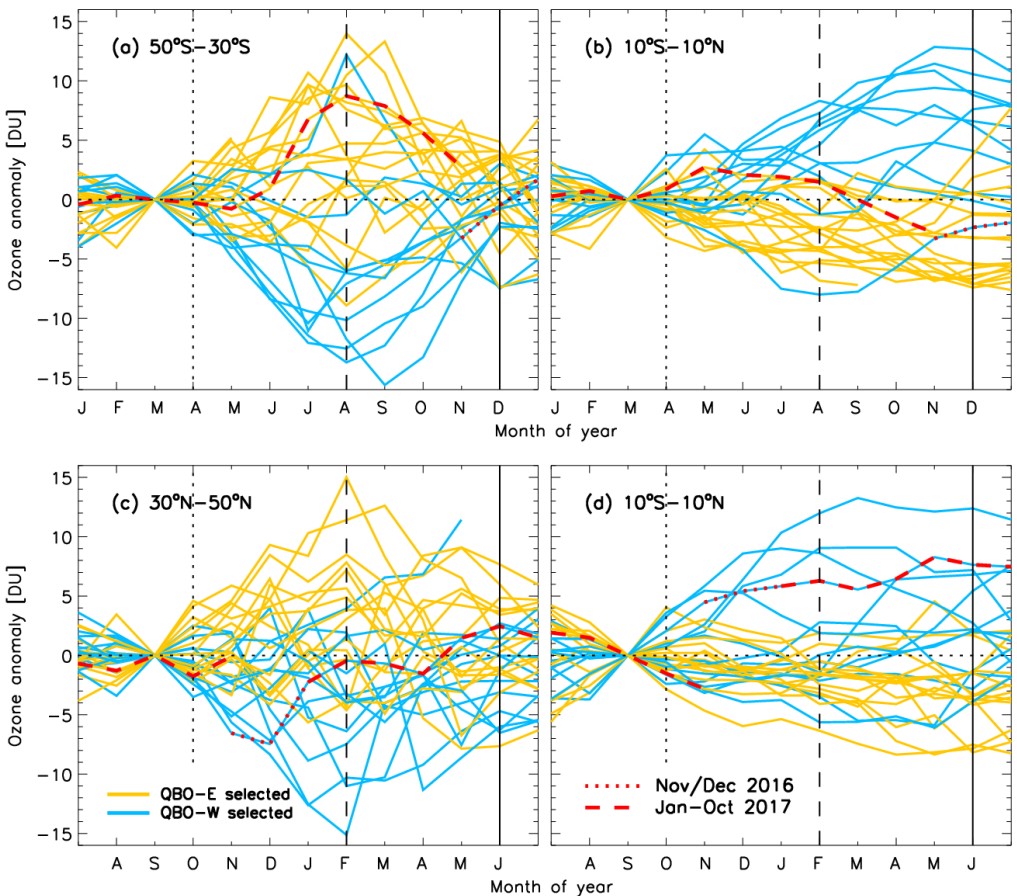

Figure 4: Lower stratospheric partial column ozone at (a) 50°–30°S, (b,d) 10°S–10°N, and (c) 30°–50°N. Each line represents a 13 month period starting in January (upper row) or July (lower row), all bias-shifted to zero in March (upper) or September (lower) and colour coded by the state of the QBO at 15 hPa in April (upper) or October (lower) so that QBO Easterly phases are yellow and Westerly blue. The period covering November to December 2016 is highlighted as a dotted-red line, while January to November 2017 is dashed-red.

The 2017 increase is highlighted in Fig. 4, with November 2016 to January 2017 shown as a dotted red line, and January to October 2017 as a red dashed line. Focusing on Fig. 4a in the SH, the increase onset during the Easterly phase is large, but as noted earlier larger excursions have occurred before and regularly (Fig. 2). A prolonged Westerly phase, following the breakdown of the expected QBO pattern in 2016 (Osprey et al., 2016; Newman et al., 2016; Tweedy et al., 2017), may have contributed to a suppressed level of ozone in 2016 (note the single orange dot in 2016 in Fig. 2a signifying a brief Easterly QBO phase). The arrival of the Easterly phase proper in 2017 led to the ozone increase at mid-latitudes. The Westerly phase at 30 hPa began in late 2018 and ozone should,

Table 1: Absolute change between 1998 and 2018 in Dobson Units, and probability (in brackets) of a positive or negative change in ozone (%) for integrated regions of the stratosphere. Blue text indicates ozone changes are negative, while red indicate positive changes; bold text indicates probabilities exceeding 90%.

| Region | 60°–30°S | 50°–30°S | 30°S–30°N | 20°S–20°N | 30°–50°N | 30°–60°N | 50°S–50°N | 60°S–60°N |
|--------|----------|----------|-----------|-----------|----------|----------|-----------|-----------|
| Whole | -0.5 (57) | -0.4 (56) | **-1.9 (95)** | **-2.5 (95)** | -3.2 (83) | -2.6 (77) | -1.1 (86) | -1.1 (86) |
| Upper | **+0.8 (100)** | **+0.8 (99)** | **+0.7 (96)** | +0.6 (85) | **+0.6 (92)** | **+0.6 (96)** | **+0.8 (100)** | **+0.8 (100)** |
| Middle | +0.7 (78) | +0.8 (78) | **-0.7 (94)** | **-0.9 (95)** | -0.5 (73) | -0.5 (72) | -0.4 (81) | -0.2 (73) |
| Lower | -1.6 (80) | -1.7 (82) | **-2.1 (99)** | **-2.1 (98)** | -1.9 (82) | -1.5 (76) | **-1.8 (99)** | **-1.7 (99)** |

barring no further QBO breakdown, decrease again in 2019 in the SH mid-latitudes; the last three months of 2018 hint at such a decrease (Fig. 2).

Despite this variability, Fig. 1 indicates that the lower stratospheric negative trends in ozone could already be identified throughout the lower stratosphere before, and after, 2016. As such, the QBO breakdown event is likely not the primary cause of the negative ozone trends reported by Ball et al.

(2018), but does appear to affect the robustness of the trend depending on the end year. We will investigate this end-year sensitivity in section 3.5.

### 3.4 Latitude-integrated lower stratospheric ozone trend estimates

While Chipperfield et al. (2018) applied ordinary least squares trend fits to timeseries using a single linear trend, this cannot be compared to multi-variate regression approaches, e.g. DLM and MLR.

This is because the former simply asks what the trend in the data is, regardless of the forcing agents, while the latter attempts to separate known (usually quasi-periodic) drivers to distill out the trend that has (usually unknown) drivers of its own. The DLM non-linear trend estimates presented here are the first multivariate analysis applied to ozone timeseries that include the large ozone increase witnessed in 2017. It is important to be clear that long-term trends cannot be compared with single

355 year changes and, indeed, the processes governing each timescale are likely quite different. While large short-term increases will likely bias the whole trend-line for that period under MLR analyses (with piecewise linear and independent linear trends – PWLT or ILT), DLM promises to be more robust in the sense that asymmetric fluctuations will only influence part of the smooth trend over a timescale fixed by the smoothness parameter $\sigma_{trend}$ that controls how rapidly the trend is allowed to

360 evolve (see Ball et al. (2017) and Fig. S1).

The DLM trends in lower stratospheric ozone estimated over 1985–2018 in Fig. 2 continue to be negative, monotonic trends up to 2018 in the quasi-global, tropical and NH regions, while the SH trend reaches a minimum in ~2013 before slowly rising. All integrated regions suggest that mean

ozone remains below the 1998 level (see Table 1[3]), though the probability of an overall decrease is 99% in quasi-global ozone, dominated by the tropics (99%), with probabilities of a decrease of 80% and 76% in the SH and NH respectively. Except in the SH, monotonic downward trends remain, with the addition of two years only affecting the gradient of the monotonic trends (compare the dotted line for the year ending in 2016, with the dashed for 2017 and solid for 2018 in Fig. 2). This agrees with Chipperfield et al. (2018) who suggested the large rapid increase of 2017 affected trends, although this was mainly in the SH and has subsequently showed little change over 2018. However, it has done little to reduce the overall probability of a decrease in the quasi-global timeseries (99%). Furthermore, the shape of the DLM curve is affected only near the end years, such that the period away from the end-date is relatively insensitive to a change in the end year and becomes 'locked-in'[4]. This is a good example of the inadequacy of using linear trends to describe these data. As the DLM-estimated changes in ozone relative to 1998 in years prior to 2010 are essentially unaffected by the addition of 2017 and 2018, the data show robustly that lower stratospheric ozone did continue to decrease until at least 2010 in all regions. We speculate that the shift back to a QBO-Westerly phase will again decrease ozone at mid-latitudes in 2019 (which appears to have begun in October 2018, see Fig. 2). If that happens, it is therefore possible that the non-linear trend estimates will likely decrease again, and the emergent 2013 minimum in the DLM non-linear trend estimate seen in Fig. 2b is likely to shift to a later date or disappear.

### 3.5 Sensitivity of DLM trends to the end year and non-linear seasonal-QBO effects

Since mid-latitude ozone excursions depend on the QBO-seasonal interaction, i.e. the QBO phase relative to the time of year, this is a non-linear mode of variability. Without predictors to represent this non-linear behaviour, linear regression models (including both DLM and MLR) cannot capture these excursions and the variability can leak into, and bias, other predictors. Most importantly, this bias may include the trend component of the regression model. In this section we examine the magnitude of this effect through a sensitivity analysis of the DLM trends to the end date of the data, both spatially and with partial columns.

Due to the magnitude of the mid-latitude, seasonally-dependent QBO ozone variability on short (two to three year) timescales, ILT or PWLT applied to the relatively short post-1997 timeseries will be sensitive to these large swings in ozone. For the smooth DLM trends on the other hand, we expect the last few years of the curve will be primarily affected, with the rest of the trend being stable. We demonstrate the impact on DLM in Fig. 5 where the partial column regions as presented in Ball et al. (2018) are shown for 10° bands, and quasi-globally (right column), over the whole stratospheric column (top), as well as upper, middle and lower. We show DLM curves estimated from six periods

---

[3]an extended supplementary materials Table S1 provides changes in DU, %, and % per decade for 1985–2018, 1985–1997, and 1998–2018.

[4]In contrast, for MLR analyses, the entire trend-line is impacted by changes in the end year (Frith et al., 2014; Weber et al., 2018; Keeble et al., 2018)

that start in 1985 and end in 2013–2018, as in Fig. 1. All curves are bias shifted to zero in 1998. This provides a visualisation of the sensitivity of the non-linear trends to the end year (and hence also the large resurgence in ozone in 2017). The uncertainties associated with a change between 1998 and

the end year are presented in Fig. 6 with 95% (dark grey shading) and 99% credible intervals. The results specifically for the 1998–2018 change are combined and presented in Fig. 7 as probability distributions, in the same manner as in Fig. 2 of Ball et al. (2018), where blue and red colours represent negative and positive changes respectively, and numbers above each distribution are the probability of the change (fraction of the probability distributions) being negative.

From the panels of Fig. 5, it is clear that ozone trends in the middle-stratosphere exhibit the largest sensitivity to the end year, and the uncertainties in the change from 1998 are consistently large (Fig. 6). Quasi-globally the middle stratosphere change since 1998 is negative for all end years, but does not exceed 95% probability. The upper stratosphere is also sensitive to the end year in the tropics (Fig. 5), and the end year shifts the estimated ozone change from negative to positive

with increasing end year, although the uncertainty always remains large (Fig. 6). At mid-latitudes uncertainties in the change of upper stratospheric ozone since 1998 are smaller, but there has been a general shift towards more positive and significant increases, which is more-clearly reflected in the SH and quasi-global estimates.

The evolution of the lower and whole stratospheric non-linear ozone trends mimic each other.

South of 30°S, the end points of the negative changes have quickly increased in 2017 and 2018 relative to 1998, though remain negative in the lower stratosphere. At latitudes north of 30°S, the addition of 2017 and 2018 has made little difference to the monotonic ozone decline and for 50–60°N, where changes are flat, the additional years make little difference. The quasi-global lower stratospheric ozone continues to exhibit a monotonic decline that is still highly confident with 99% probability

(Fig. 7 and Table 1), and ozone abundances integrated over the whole stratosphere continues to remain lower in 2018 than in 1998, though this is now with a probability of 86%; these trends are dominated by the tropical contribution (58%, latitude weighted) to the quasi-global change, whereas the NH and SH contribute 21% each. Even so, the NH changes do not appear affected by the recent large seasonally-dependent QBO variability.

Figure 5 also confirms that the gradients of the non-linear curves are only affected by unmodelled variance in years close to the end points, typically within the last five years of the partial column timeseries considered here. The shape of the DLM curves prior to the final five years of the DLM curves are largely unaffected. Indeed, even with the large ozone increase in 2017 in the SH, we see that all trend-curves agree well prior to 2010. This is also true in other panels, e.g. the middle

stratosphere and tropical upper stratosphere. In the upper stratosphere the recovery onset remains robust, but in the SH lower stratosphere the large increase in 2017 results in the non-linear trend curve having a local minimum emerge around 2013. As such we can infer that additional data are unlikely to affect the inferred change of ozone in 2013, relative to 1998, or push the minima to earlier

dates, because the affecting end year moves further away with more data. However, subsequent data
might once again push the changes since 1998 to lower levels, e.g. if mid-latitudes do respond to
a Westerly-phase QBO with ozone reducing sharply as it has done in the past (Fig. 2). We expand
the idea of inferring the likely earliest minimum using the DLM with spatially-resolved data in the
Supplementary Materials.

### 3.6   Update on ozone profiles

Briefly, in Fig. 8, we provide updated ozone change profiles for 1998–2018 using the standard latitu-
dinal ranges for the SH (60°–35°S), tropics (20°S–20°N), and NH (35°–60°N) (WMO, 2014, 2018;
Steinbrecht et al., 2017; Petropavlovskikh et al., 2019). Fig. 8, also includes 1998–2016 and 1998–
2017 profiles for comparison and shows that, for 1998–2018, confidence in an upper stratospheric
ozone recovery from ODSs is clear for all latitude bands, including the tropics where it has pre-
viously remained below the 95% significance levels. The lower stratosphere shows negative ozone
changes at almost all levels, though these generally do not exceed a probability of 95%.

Ball et al. (2018), and Fig. 7, indicate that the 50–60° zonal means in both hemispheres show
little ozone change in the lower stratosphere in the last 21 years, while the tropical regions out
to 30° show a strong decrease. By modifying the latitudinal extent of the profiles slightly, so that
mid-latitudes cover 30–50° to exclude 50–60° and the tropics are widened to 30°S–30°N to include
the subtropics, the modified profiles are presented in Fig. 9. This provides some measure of the
sensitivity to the latitudinal ranges chosen. Now we see the tropics show close to 95% confidence of
an ozone decrease at all tropical lower stratospheric pressure levels, and there is increased confidence
of an ozone reduction in the mid-latitude lower stratosphere. Further, the inclusion of higher latitude
regions (20–30°) reinforces the tropical upper stratosphere ozone increase.

An upper stratospheric increase is the expected result from long-term stratospheric chlorine re-
ductions, a direct consequence of the Montreal Protocol and its amendments, though we do not
explicitly attribute the cause of the increase to that here (for more on attribution see, for example,
WMO (2018)). Indeed, the Montreal Protocol and its amendments will have been effective in re-
ducing ozone losses throughout the atmosphere through reductions in CFC emissions, HCFCs and
other ODSs. The lack of a positive trend since 1998 in the lower stratosphere, as opposed to the one
clear in the upper stratosphere, is likely the consequence of other factors such as dynamical changes
(Wargan et al., 2018). These results once again reinforce the conclusion that only the SH is affected
by the 2017 ozone increase (lower stratosphere), that the Montreal Protocol appears to be working
(upper stratosphere), and that the decreases in the lower stratosphere at tropical and NH latitudes
remain in place, but are not yet fully understood.

## 4  Conclusions

Here, we have extended and analysed the BASIC$_{SG}$ stratospheric ozone composite from Ball et al. (2018) by two years to cover 1985–2018. BASIC$_{SG}$ merges two composites, SWOOSH and GOZ-CARDS. We perform a set of sensitivity tests, using dynamical linear modelling (DLM), on the post-1997 trend estimates to understand the impact of a recently reported, large increase in modelled ozone in the lower stratosphere in 2017 (Chipperfield et al., 2018), following almost two decades of persistently decreasing ozone.

The aim of this work is to assess the current state of, and trends in, stratospheric ozone. Improved knowledge of such trends, and the relevant forcing mechanisms and associated variability, will help to better constrain CCM projections of ozone to the end of the 21$^{st}$ Century. Chemistry models resolving the stratosphere are one of the best tools for attribution and long-range studies of ozone, but different types exist. Free-running CCMs generate their own model-dependent internal climate and variability. Chemistry transport models (CTMs) use wind, temperature and surface pressure fields fully prescribed by reanalyses. And, specified-dynamics CCMs (SD-CCMs) use reanalyses to nudge the internally-generated variability of the model closer to the historical variability in the real atmosphere while attempting to retain model dependent processes and internal consistency. CTMs and SD-CCMs can be useful for attributing historical changes in ozone to evolving concentrations of $CO_2$ and ODSs (Solomon et al., 2016), or the Sun (Ball et al., 2016), by accounting for dynamical variability in observations.

A recent study (Chipperfield et al., 2018) used a CTM to reconstruct the ozone timeseries beyond the observational record available at the time to 2017 and found that that model simulated a lower stratospheric ozone increase in 2017 back to 1998 levels. This increase was attributed to dynamical variability. Indeed, chemistry and photochemistry play a dominant role over dynamical perturbations in the upper stratosphere as ozone lifetimes are short (~days), while ozone lifetimes of ~6–12 months in the lower stratosphere means that equator-to-mid-latitude transport of similar timescales plays an important (dominant) role there (London, 1980; Perliski et al., 1989; Brasseur and Solomon, 2005). CTMs can provide insight as to whether the changes might be driven by photochemistry, chemistry, or dynamics. However, because the dynamical fields are prescribed, the CTM cannot provide insight into the underlying dynamical driver of the long-term decreases or the 2017 increase. We show here that the 2017 increase simulated by the CTM (Chipperfield et al., 2018) was more than 60% larger than that observed, and that the 1998-2017 and -2018 (Fig. 1e and f) change remains negative (60°S-60°N), and significant in the tropics and some sub-regions of the NH (Fig. 1f). Neither free-running CCMs (WMO, 2014), nor SD-CCMs (Ball et al., 2018), have so far been demonstrated to accurately reproduce the long-term changes estimated from observations in lower stratospheric ozone (Fig. 6).

The effect of the ozone increase in 2017 was small and the probability of an overall ozone decrease in the lower stratosphere remains at 99% for 1998–2018 (-1.7 DU, or 2.0%; see Table S1). We note that the lower stratospheric ozone trends are dominated by the tropical regions (30°S–30°N) where

the decrease is robust to the end year over 2013–2018, with a probability of 99% (-2.1 DU, -3.5%)
that it was lower in 2018 than in 1998. Nevertheless, mid-latitudes out to 50°N also indicate that the
decrease persists (-1.9 DU, -1.7%). We also find that the 2017–2018 addition enhances the estimated
magnitude of the upper stratospheric ozone positive trend, but that the quasi-global (60°S–60°N)
ozone layer still displays a reduction since 1998, though the confidence in this has reduced from 95%
in 2016 (Ball et al., 2018) to 86% in 2018 (-1.1 DU, -0.4%). Given the high probability of a decrease
in tropical middle (94%) and lower (99%) stratospheric ozone, the whole tropical stratospheric ozone
column indicates a highly probable decrease (95%) over 1998–2018 (-1.9 DU, -0.8%).

In general, uncertainties on changes since 1998 in partial columns have changed little over 2013–
2018 (Fig. 6), a result likely due to the large fraction of unaccounted variance in the standard set of
predictors used in regression analysis. Our analysis shows that ozone continued to decrease until a
minimum in at least 2013 in the SH, and has continued to decrease at all latitudes north of 30°S. By
comparing the phase of the QBO with large, 2–3 year inter-annual variability at mid-latitudes, the
implication is that these large mid-latitude changes are related to the seasonal-dependence of ozone
on the QBO, i.e. a non-linearity. If true, this could explain why regression models cannot capture this
variability, since such non-linear behaviour is not included. The clarification of the origin of these
large mid-latitude changes – occurring every few years – is a high priority.

CCMs are consistent in the sign of their projections, although lower stratospheric ozone variabil-
ity can differ with observations and there is a large spread in their sensitivity to hODSs (Douglass
et al., 2012, 2014), and therefore their return dates, i.e. a return of ozone to the level it was in 1980
(WMO, 2014; Dhomse et al., 2018; WMO, 2018). CCMs do a good job on many timescales, but
due to historically different internal variability, and parametrized sub-grid scale processes and nu-
merical diffusion, behaviour in some regions may not be well-reproduced (SPARC/WMO, 2010).
It is clear from modelling studies that pre-Montreal Protocol ozone decreases can be attributed to
ODS increases (WMO, 2014), and SD-CCMs and CTMs generally reproduce the Antarctic ozone
hole well (Solomon et al., 2016). The halt in ODS-related ozone losses as a result of the Montreal
Protocol and its amendments, and an initial recovery from ODSs in total column ozone is almost
universally reproduced by CCMs (SPARC/WMO, 2010), as is the upper stratospheric ozone recov-
ery. But, negative ozone trends since 1998 in the lower stratosphere have not been demonstrated to
be simulated in models in the mid-latitudes, most notably in the NH.

Future projections tend to focus on how stratospheric ozone will evolve under a given global
warming scenario. This is important given that anthropogenic GHG emissions that are changing the
climate may impact inter-annual dynamical variability in the stratosphere (Osprey et al., 2016; New-
man et al., 2016; Tweedy et al., 2017). Changes are also expected in the large-scale circulation of the
stratosphere, and these are likely to modify future distributions of ozone (Chipperfield et al., 2017).
Further, ozone is not a passive tracer, and the large scale long-term changes in ozone are expected
to feedback on the aforementioned dynamics (Li et al., 2018; Polvani et al., 2018; Abalos et al.,

2019). Such a feedback has been demonstrated, most notably, in the SH following ozone depletion and the growth of the ozone hole (WMO, 2014, 2018). Now, as ozone is expected to recover in the coming decades, the dynamics of the stratosphere are also expected to respond. The overall future expectations are that total column ozone levels will return to 1980s levels globally by ~2050, in the

545 Antarctic by 2100, and by ~2030 and ~2050 in Northern and Southern mid-latitudes, respectively. The mid-latitudes are expected to continue on to a 'super-recovery', i.e. that ozone will be higher by the end of the 21$^{st}$ Century than prior to 1980s levels (Dhomse et al., 2018; WMO, 2014, 2018), although this is predicated on future scenarios of hODSs decreases continuing as expected (Montzka et al., 2018). However, it is neither clear whether the recent increase in SH lower stratospheric ozone

will remain at higher levels or will reduce again in 2019 as the QBO shifts to a Westerly phase, nor why the NH continues to show a persistent decrease. Nonetheless, we note that the signal is small compared to the (i) large inter-annual variability, (ii) pre-2000 changes induced by ozone depleting substances, and (iii) ozone loses that would have occurred without the Montreal Protocol being enacted.

The ongoing negative trend of ozone in the lower stratospheric component of the total column also continues to pose a problem for global trends in tropospheric ozone. If tropospheric ozone has really increased over the last two decades, and stratospheric ozone was not decreasing or remained flat, then some component of the total column ozone must have been decreasing to balance the ozone budget since it appears that total column ozone has remained steady in the past 5-10 years.

Alternatively, it is possible that the solution simply lies in very large observational uncertainties (Harris et al., 2015; Gaudel et al., 2018; Petropavlovskikh et al., 2019) and/or the inadequacies of linear regression techniques to attribute variability and identify trends. In addition to potential future improvements in merged observational records, this calls for a community push to improve detection and attribution techniques to solve an issue that is of great importance to the health of society, the

biosphere, and the climate.

*Author contributions.* WTB designed the experiments. WTB and JA prepared and executed the BASIC algorithms. JA developed the DLM code and WTB and JA performed the DLM analysis. WTB conceived and performed the QBO analysis. SD and LF prepared and provided GOZCARDS and SWOOSH ozone datasets. WTB prepared the manuscript with contributions from all co-authors.

*Acknowledgements.* W.T.B. was funded by the SNSF project 200020_182239 (POLE). 'BASIC$_{SG}$' for 1985–2018 will be available for download from https://data.mendeley.com/datasets/2mgx2xzzpk/3 following review of this manuscript. Work at the Jet Propulsion Laboratory was performed under contract with the National Aeronautics and Space Administration. GOZCARDS ozone data contributions from Ryan Fuller (at JPL) are gratefully acknowledged.

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

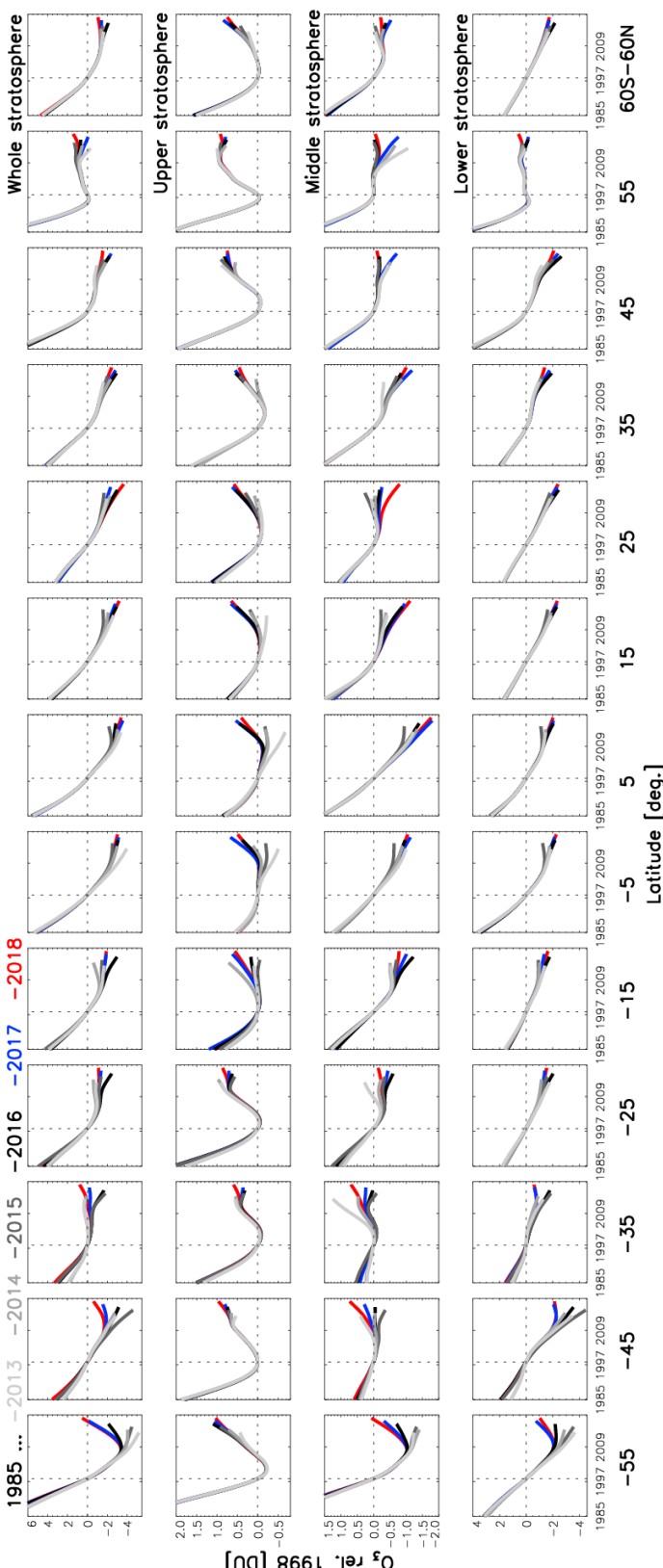

Figure 5: The partial column ozone non-linear trends estimated as a function of end year (2013 to 2018; dark to light colours), for each 10° latitude and quasi-global (left to right) and (top to bottom) the whole, upper, middle and lower stratosphere. Each sub-panel covers 1985–2018 and all curves are bias corrected to January 1998 (horizontal and vertical dotted lines). Uncertainties for each 1998–end-year change are given in Fig. 6.

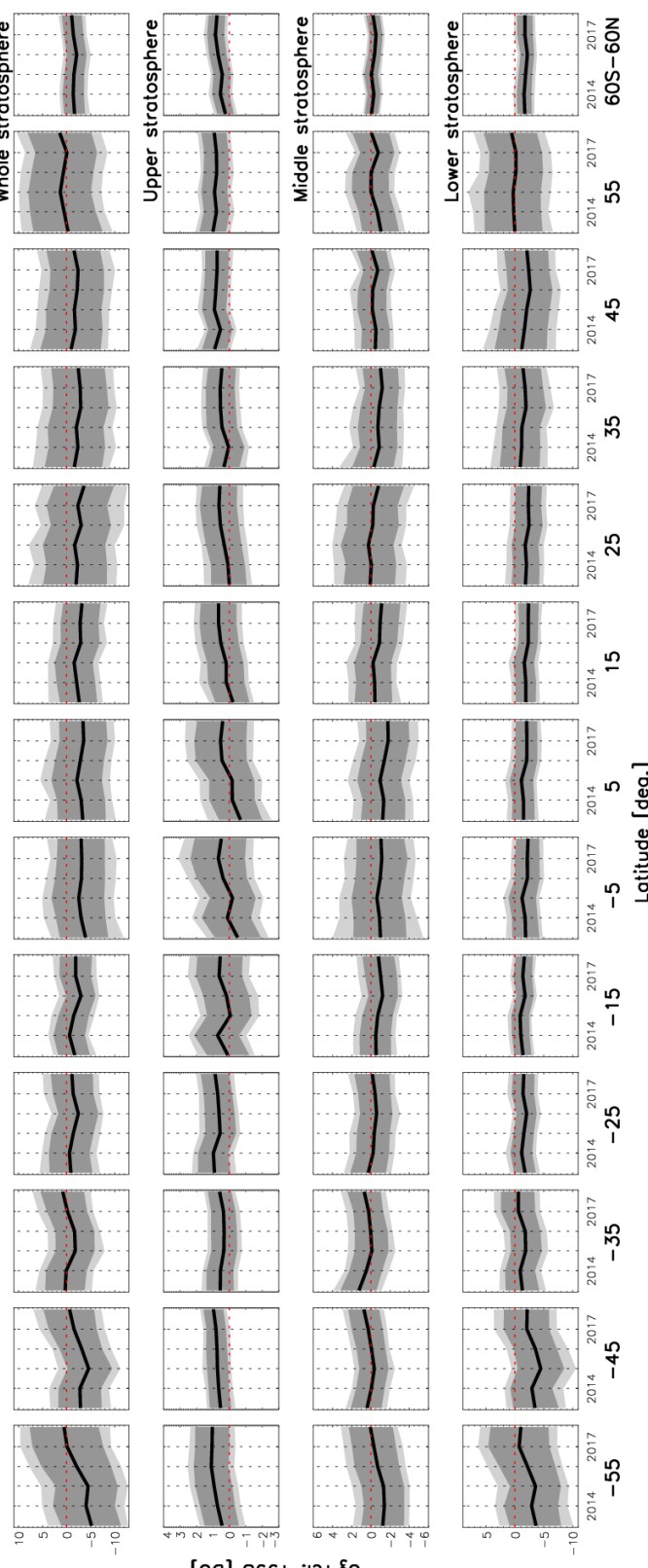

Figure 6: The partial column ozone changes between 1998 and the end-year from 2013 to 2018 (x-axis of each sub-panel) from the non-linear trends as in Fig. 5. Dark and light shading represent 95% and 99% credible intervals.

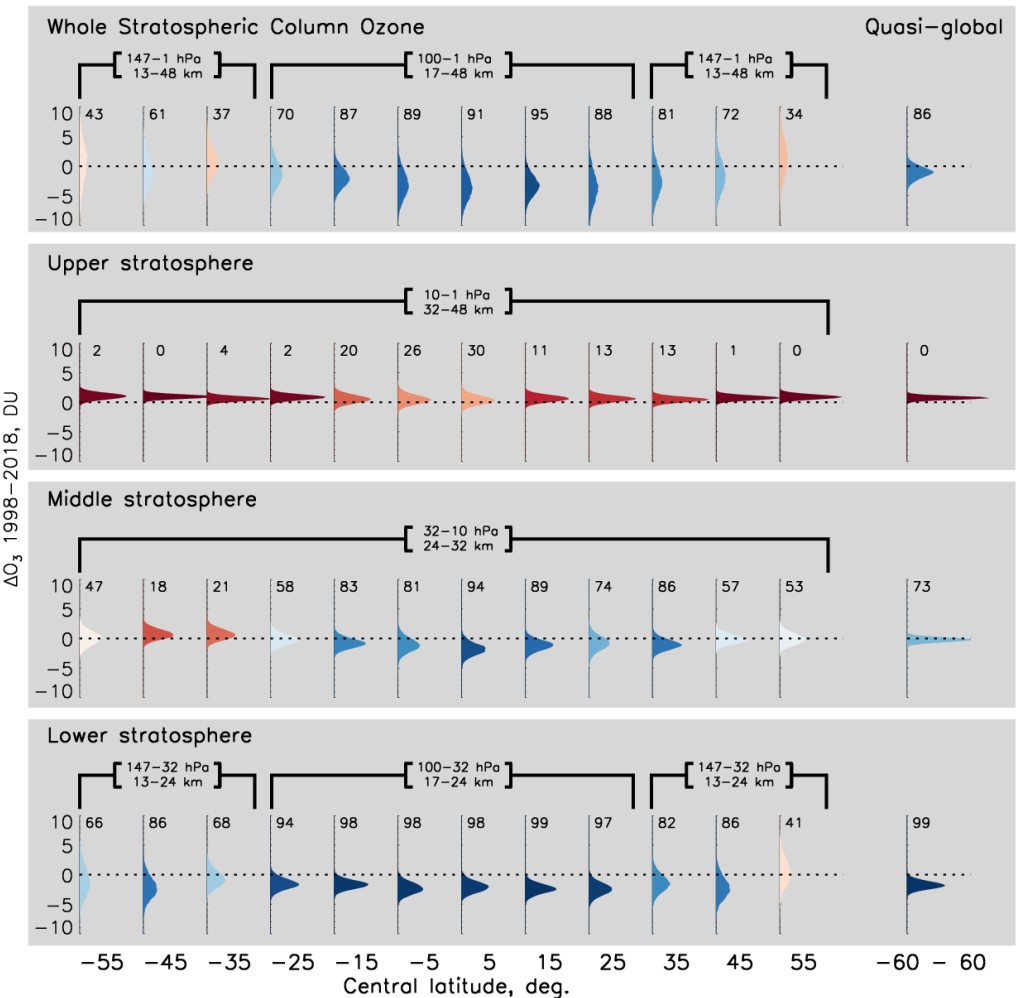

Figure 7: Posterior distributions (shaded) for the 1998–2018 partial column ozone changes. (Top) whole stratospheric column, (middle) upper and (bottom) lower stratosphere in 10° bands for all latitudes (left) and integrated from 60°S–60°N ('Quasi-global', right). The stratosphere extends deeper at mid-latitudes than equatorial (marked above each latitude). Numbers above each distribution represents the distribution-percentage that is negative; colours are graded relative to the percentage-distribution (positive, red-hues, with values <50; negative, blue).

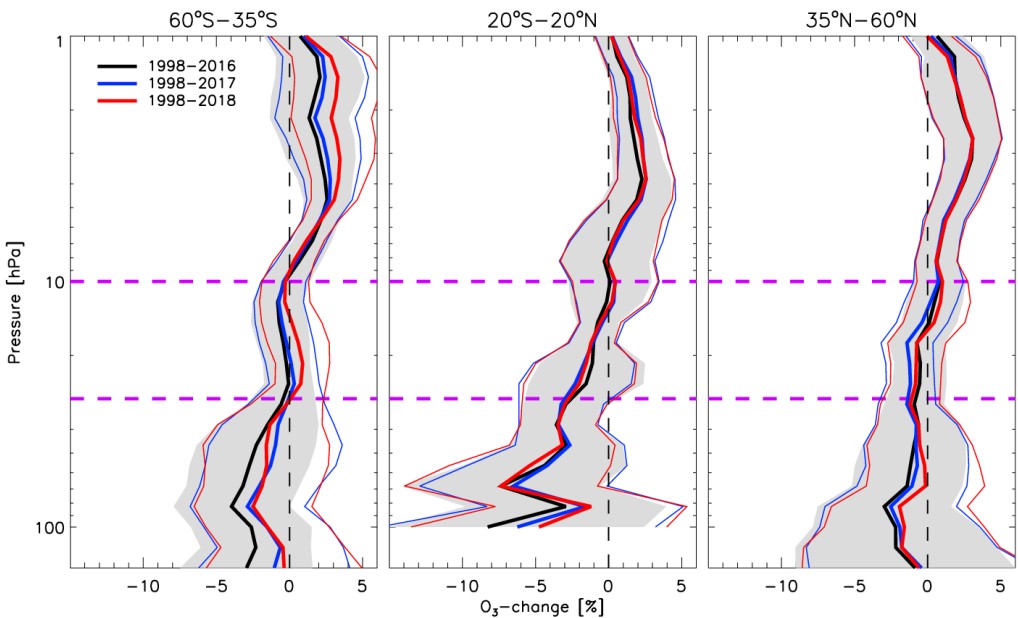

Figure 8: The ozone profiles for 1998 to an end-year of 2016 through 2018 (see legend) in the southern hemisphere (60°–35°S), the tropics (20°S–20°N), and the northern hemisphere (35°–60°N). Shading is for 2016 only. Uncertainties are 95% credible intervals. Pink lines indicate boundaries of partial columns in other figures.

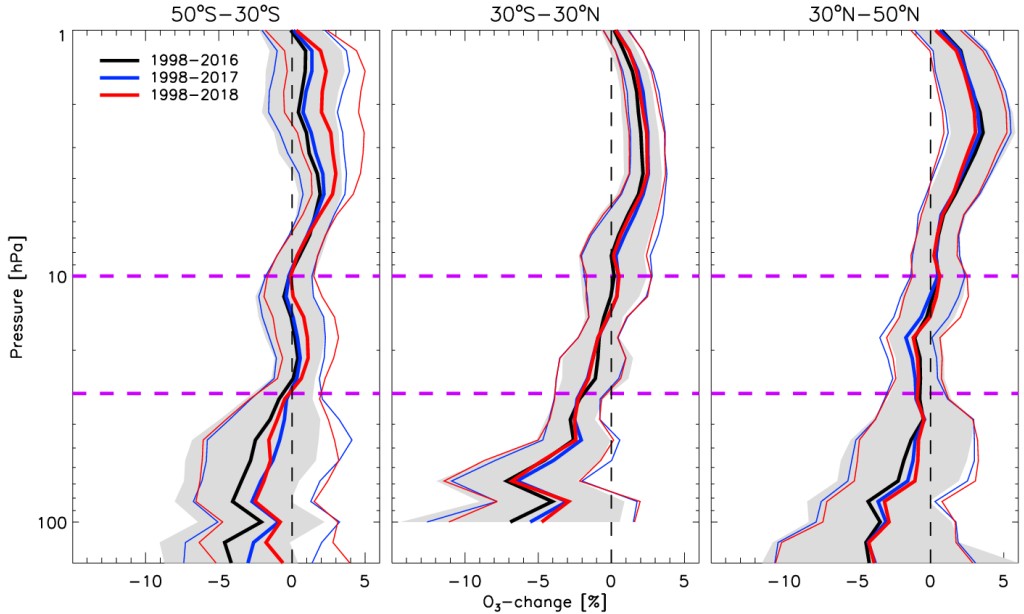

Figure 9: As for Fig. 8, but for the 50°S–30°S (SH), 30°S–30°N (tropics), 30°N–50°N (NH).