# Peer review of "Stratospheric ozone trends for 1985–2018: sensitivity to recent large variability"

_Atmospheric Chemistry and Physics, 2019_

## Short Comment (SC1) · 26 Mar 2019

Two paper by Diallo et al. (2018) and (2019) recently published in ACP discusses the combined influence of QBO and ENSO on the UTLS ozone and water vapour distributions using a lagged multiple regression analysis. A reference to these paper should be included along with a discussion of the added scientific value of the results presented here.

The papers are accessible here:

1. Diallo, M., Riese, M., Birner, T., Konopka, P., Müller, R., Hegglin, M. I., Santee, M. L., Baldwin, M., Legras, B., and Ploeger, F.: Response of stratospheric water vapor and ozone to the unusual timing of El Niño and the QBO disruption in 2015–2016, Atmos.

Chem. Phys., 18, 13055-13073, https://doi.org/10.5194/acp-18-13055-2018, 2018.

2. Diallo, M., Konopka, P., Santee, M. L., Müller, R., Tao, M., Walker, K. A., Legras, B., Riese, M., Ern, M., and Ploeger, F.: Structural changes in the shallow and transition branch of the Brewer–Dobson circulation induced by El Niño, Atmos. Chem. Phys., 19, 425-446, https://doi.org/10.5194/acp-19-425-2019, 2019.

---

## Referee Comment (RC1) · Anonymous Referee #1 · 17 Apr 2019

This manuscript is an update and extension of a similar analysis by Ball et al. (ACP, 2018), which reported evidence from satellite data that ozone in the lower stratosphere at latitudes less than 60 degrees has continued to decline since 1998 even though upper stratospheric ozone has started to recover in response to the Montreal Protocol bans of many ozone-destroying substances. Since much more ozone is in the lower stratosphere than in the upper stratosphere, the lower stratospheric decline dominates. On the other hand, evidence for an increase in tropospheric ozone was reported that could potentially cancel out the lower stratospheric decline. In that 2018 paper, no conclusion about the cause of the continued lower stratospheric decline was drawn and possible explanations involving both dynamics and chemistry were only briefly discussed.

In this manuscript, the trend analysis is slightly updated by the addition of more data and a stronger conclusion is drawn that the cause of the continued decline in the lower stratosphere is dynamical in nature rather than chemical. Specifically, the manuscript agrees with a recent study by Chipperfield et al. (GRL, 2018) that a strong positive ozone anomaly in 2017 was driven by short-term dynamical transport of ozone. However, it is further shown here that this short-term increase was caused by the stratospheric quasi-biennial oscillation and that a long-term gradual decline in lower stratospheric ozone remains even when the QBO-related variability is taken into account. The manuscript also agrees with the Chipperfield et al. conclusion that short-lived chemical depletion cannot explain the long-term decline, which must therefore be dynamical in origin.

I have two main comments on the manuscript, at least one of which can be considered as major. If the authors can satisfactorily address these comments in their revision, publication can be recommended.

(1) The magnitude of the long-term trends in zonal mean extrapolar lower stratospheric ozone is not clearly stated in the abstract or in most parts of the text. This leaves readers in the dark about how important the declines are. According to Figure 2a, the overall quasi-global decrease over 1985-2018 (33 years) is roughly 3.5 DU, which is in the range of 1.0 to 1.5%, or roughly -0.5% per decade. For comparison, the increase in tropospheric column ozone in the same latitude range estimated by Ball et al. (2018; their Figure 4) is +1.68 +/- 0.11 DU per decade or roughly +0.5% per decade. Could the authors please add numbers like this to the abstract and conclusions sections? The present manuscript notes large uncertainties in the actual tropospheric ozone increase (lines 38-41). However, at least the magnitude of the estimated lower stratospheric ozone decreases and the strong possibility of compensating tropospheric ozone increases should be more explicitly stated. Then readers can judge for themselves the importance of the observed lower stratospheric ozone decline due to dynamical processes.

(2) In this manuscript, as well as in the published work of Ball et al. (2018), the seasonal and longitude dependences of the observed zonal mean ozone trends are not evaluated. By not evaluating these characteristics or at least reviewing previous work on these characteristics, the authors are missing some important clues for understanding the origin of the observed zonal mean ozone trends in the lower stratosphere. Beginning in the 1990's, a number of authors have found that there is both a seasonal and a longitude dependence of column ozone trends, most of which originates in the lower stratosphere, especially at middle latitudes in winter and spring. Some of these authors concluded that these dependences are a consequence of decadal variability of quasi-stationary ultra-long Rossby waves that propagate from the troposphere into the stratosphere (Hood and Zaff, JGR, 1995; Peters and Entzian, Meteorol. Z., 1996; Peters et al., Beitr. Phys. Atmos., 1996; Hood et al., JGR, 1997; Hood et al., JGR, 1999). Decadal climate variability was therefore implicated. This was an extremely controversial and unpopular conclusion at the time because the prevailing view was that the observed ozone trends in both the polar and extrapolar regions in the lower stratosphere were dominantly chemical in origin. Note that these quasi-stationary waves are not necessarily linear (even though most mathematical treatments make this assumption) so that the zonal mean ozone change is not necessarily zero. Moreover, there is an associated trend in the BDC, which also affects ozone amounts, because the BDC is driven by Rossby wave breaking and absorption in the stratosphere (e.g., Hood and Soukharev, JAS, 2005). The continued decline of lower stratospheric ozone reported in this manuscript may therefore imply that there is a long-term component of the variability of these ultra-long waves. Schneidereit and Peters (Atmosphere, v. 9, p. 468, 2018) have recently investigated the zonally asymmetric component of ozone trends over 1979-2016 and find negative trends over Europe in winter that are twice as large as the zonal mean trend. A strong negative trend in one area such as this could produce a net change in the zonal mean ozone if the waves are non-linear. These authors suggest that long-term changes in the Rossby wave train that propagates out of the tropics and is linked to Arctic warming may be the ultimate driver. If the authors

are not able to investigate zonally asymmetric ozone trends or long-term changes in quasi-stationary waves and their breaking behavior in the stratosphere for the present manuscript, then they should at least include some discussion of this previous work with references to some of the above papers.

---

## Referee Comment (RC2) · Anonymous Referee #2 · 22 Apr 2019

**GENERAL COMMENTS**

This paper presents a compelling update to the earlier work of Ball et al., and a much awaited response to the Chipperfield paper. While the science is excellent, unfortunately the writing is often too curt and the reader has to guess at what is being referred to (see many examples below). The writing, in many places, indicates a lack of clarity of thought, or perhaps just that the author is not writing what he means to say. I would strongly recommend that the author improves the quality of the writing. There are no major changes required for this paper to be ready for publication. Most of my suggested changes below are relatively minor. It is an excellent piece of analysis. Now it just needs to be communicated to its audience clearly.

**SPECIFIC COMMENTS**

[Figure]

Line 1: I would suggest replacing 'The Montreal Protocol' with 'The Montreal Protocol, together with its amendments and adjustments'.

Line 3: I would suggest that you reserve the word 'recovery' strictly to talk about the recovery of ozone from the effects of ODSs. When you are talking more generally about ozone increases, whether driven chemically or dynamically, talk rather about 'ozone increases' rather than 'ozone recovery'.

Line 12: I am struggling a bit with this 'still likely lower than in 1998 (probability $\sim$80%)'. Surely it is either lower or it isn't. OK , maybe I will get to see later how this is nuanced by some statistical significance.

Lines 16-17: With regard to the sentence 'These decreases do not reveal an inefficacy of the Montreal Protocol', an important point that can, and perhaps should, be made here is that tropical stratospheric ozone is almost certainly recovering (from the effects of ODS), and hence the Montreal Protocol is successful, while simultaneously still declining (due to other influences). The point to make is that recovery and declining ozone are not mutually exclusive.

Line 33: You need to be clear here what you mean by 'significant'. Do you mean statistically significantly different from zero at the 2 sigma level, or do you mean a more general 'significant' as in 'large'. It matters a lot in this specific context so I think that you should be clear.

line 59: I think it would be plausible to say that climate change may be exacerbating some specific dynamical mechanism (or more than one) that affects ozone, but it feels incongruous to state that climate change, in and of itself, could be a mechanism for dynamically affecting ozone.

Line 125: Delete 'point the reader to Laine et al. (2014) for details on this method and' since you have anyway cited the Laine et al., 2014 paper.

Line 135: This essentially assumes that the sensitivity of ozone to the regressors is

time-dependent. What support is there for this assumption? Why might ozone have a certain sensitivity to EESC in 1985 and then a different sensitivity in 1995, or 2005? This ability for DLM to accommodate changes in the sensitivity of ozone to the regressors is presented as an advantage of DLM over MLR but I am not convinced that it is. In choosing DLM over MLR you are making some significant assumptions and I am not sure that you have support for those assumptions. Or is it just the amplitude and phase of the seasonal cycle that you allow to vary with time?

Line 138-139: Well, unless EESC is selected as a descriptor of the long-term secular trend in which case it is a much more natural and appropriate descriptor than any pure statistical descriptor.

Lines 143-144: I don't be believe it is true that 'in practice MLR is often performed by first subtracting an estimated mean seasonal cycle'. I certainly don't. Most MLR-based analyses I have seen fit the annual cycle as a series of Fourier expansions along with all of the other regressors.

Line 151: What is MCMC? I have not seen this acronym defined anywhere.

Line 170: Can you give some indication of why the SAOD time series is not available beyond 2016?

Line 196: It wasn't clear to me what was meant by 'this group of spatial responses'. Responses of what to what? Are you referring the latitude/pressure resolved trends plotted in Figure 1?

Line 214: But the latitudinal extent of the changes you are seeing in observations is much wider than what is seen in the CCMs right?

Line 240: But only in the lower stratosphere right?

Line 253: I think that you need to read and cite Gray, L.J. and Pyle, J.A., A two-dimensional model of the quasi-biennial oscillation of ozone, J. Atmos. Sci., 46, 203-220, 1989. Another paper that might be relevant is Bodeker, G.E.; Garny, H.; Smale,

[Figure]

D.; Dameris, M. and Deckert, R., The 1985 Southern Hemisphere mid-latitude total column ozone anomaly, Atmos. Chem. Phys., 7, 5625–5637, 2007, especially if you are seeking clarification of the origin of the large mid-latitude changes in ozone that occur every few years.

Line 311: It is not clear to me what you mean by 'governing each other'? Do you just mean 'governing each'?

Line 325: This is worded in a very confusing way - please rewrite.

Line 329: As *what* in years prior to 2010 are essentially unaffected by the addition of 2017 and 2018? Ozone in 2013 is unaffected by the addition of 2017 and 2018? You are referring to something being unaffected by the addition of 2017 and 2018 but I am not sure what that something is.

Line 334: You need to make it clear that you are referring to the minimum in the DLM fit and not a minimum in the observed ozone.

Line 336: Do you mean mid-latitude ozone excursions depend on the phasing of the QBO phase from westerly to easterly (or vice versa) on the phase of the annual cycle in ozone? If so, please consider wording as such. If not, please reword to be more clear.

Line 337: Can't you just have cross-terms in your regression model i.e. a QBO basis function modulated by a phase dependent seasonal cycle?

Line 338: I don't think that is true. They can, they just need to include the appropriate regression model basis functions. Perhaps many MLR models currently in use do not, but that does mean that they fundamentally can't.

Line 358: Uncertainties in what are consistently large?

Line 359-360: The upper stratosphere is also sensitive to what in the tropics? Too often the explicit subject of a sentence or phrase is omitted in your writing which requires the

reader to constantly be guessing at what you are referring to. This makes deciphering the narrative very tiring. There are many examples of this in my comments. The next one is on the same line (360):

Line 360: What has 'shifted from negative to positive'. And what, exactly is uncertain?

Line 361: Uncertainties in what are smaller? And smaller compared to what?

Line 361: There has been a general shift in what towards more positive and significant increases? Please read the sentence that starts on line 359 and ends on line 362 i.e. "The upper stratosphere is also sensitive....SH and quasi-global estimates" and see whether, read as it stands, it would make sense to someone. Take that sentence to a colleague and ask them to tell you what it means. They may be horrified to read that the stratosphere has shifted from negative to positive. Maybe they always thought that the stratosphere was positive and may be alarmed that it has become negative. But at least they will know that whatever happened, its (whatever it is) is always uncertain. I would strongly suggest that you write in a way that prevents the reader from having to guess at things.

Line 367: The statement that 'The quasi-global lower stratosphere continues to exhibit a monotonic decline' is not true. There are many things in the quasi-global lower stratosphere that are not continuing to exhibit a monotonic decline. A good example would be $CO_2$. Please work to improve the precision of your writing.

Line 368: I was shocked to read that 'the whole stratosphere continues to remain lower than in 1998'! Is the sky really falling? https://www.youtube.com/watch?v=NO04VXBIS0M. Is there nothing we can do to lift the stratosphere?

Line 374: Regarding 'changes prior to the last five years are largely unaffected in the partial columns'. I would be horrified if it was possible that ozone prior to the last five years was affected by the addition of recent years. It would mean that someone,

somewhere, has invented time travel. But perhaps that's not what you mean? It might be a good idea then to write \*exactly\* what you mean.

Line 403: So are you really saying that the Montreal Protocol is working only in the upper stratosphere and not in the lower stratosphere. This will hugely concern policy-makers. They will wonder why all the hard work they have done since 1987 in reducing emissions of CFCs, halons, HCFCs and other ODSs has only decreased their concentrations in the upper stratosphere. Could I put it to you that the Montreal Protocol has been effective in reducing ODS concentrations, and thereby concentrations of Cly and Bry throughout the atmosphere, and that, as a result, ozone throughout the atmosphere, including the lower stratosphere, is recovering from the effects of those ODSs. Is this recovery apparent in observations in the upper stratosphere? Apparently yes. I say apparently only in that (at least in this paper) a thorough attribution of the drivers of those ozone increases has not been done. Is this recovery apparent in observations in the lower stratosphere? No, clearly not? Why not? Well because other factors have been affecting ozone (not diagnosed in this paper) that are likely (we cannot be sure since a thorough attribution has not been done) overwhelming the increases brought about by reductions in concentrations of Cly and Bry. Wouldn't that be a more accurate picture to communicate to policy-makers?

Line 412: What does it mean to be 'confident in the evolution of stratospheric ozone'? You're confident that ozone is evolving? I'm pretty confident it is too. It always has been. I don't need your paper for that. Maybe you mean that the aim of this work is to build confidence in our quantitative understanding of trends, and other long-term variability, in upper, middle and lower stratospheric ozone?

Line 414: I think that the best tools for studying long-term changes in ozone, and attributing the causes of those changes to known drivers, is the application of regression models to observations. Yes, models can be useful for attribution but they have little role, if any, in detecting the changes in the first place. I think that chemistry-climate models are the best tools for making projections of how ozone may change in the

future.

Lines 423-425: This sentence blurs the lines between observations and model output. Were the 2017 data from Chipperfield CTM output or observations? If it was model output I would suggest replacing 'found lower stratospheric ozone had rapidly increased in 2017 back to 1998 levels' with 'found that model simulated lower stratospheric ozone increased in 2017 back to 1998 levels'.

Line 429: Any idea why the CTM got it so wrong?

Line 438: Does it enhance the positive trend in ozone, or does it enhance the recovery of ozone from ODSs - noting that those are two very different things?

Lines 439-440: How can the 'recovery' display a 'reduction'. That makes no sense to me. I can understand how ozone can reduce. I can even understand that ozone can reduce while simultaneously recovering from the effects of ODSs (I am not saying that that's what is happening here, but it is conceivable).

Line 447: By 'continues at all latitudes north of 30°S' do you mean continues to decrease at all latitudes north of 30°S?

Line 449: The seasonal-dependence of the QBO on what? Or do you mean the seasonal dependence of ozone on the QBO?

Line 455: This is the first mention of 'return dates'. What is meant by this? What is the 'return date' in a CCM?

Lines 456-457: What, exactly, do you mean by 'numerical inaccuracies'? Can you please add a sentence or two that elucidates this.

Line 460-461: Wait a minute. I have seen no evidence anywhere that there has been a 'halt in the recovery in total column ozone' from the effects of ODSs. I have seen plenty of evidence that ozone in different regions of the atmosphere continues to decline (including this paper) but no attribution of this declines such that one could conclude that

ozone is not recovering from the effects of ODSs. In fact it would devastate our understanding of atmospheric chemistry if it was found that decreasing concentrations of Cly and Bry had no impact whatsoever on the Cly and Bry cycles that destroy ozone. I see chemists leaping from buildings. So I strongly reject your conclusion that ozone, anywhere in the atmosphere, is not recovering from the effects of ODSs as you have presented no evidence at all to that effect. To make that call, you would need to do a robust attribution of changes in ozone to date and demonstrate that the ozone changes attributed exclusively to changes in Cly and Bry have been negative. And that you have not done.

Line 464: Do you mean predictions or projections? I think that it is very dangerous to use models to make predictions.

Line 468: Do you mean total column ozone or do you mean ozone in all parts of the atmosphere?

Line 470: I have no idea what you mean by 'super-recovery'. I fully understand how ozone can recover from the effects of ODSs. But I can't understand how it can 'super-recover'. I can understand how ozone could become higher than it was in the 1960s, but this has nothing to do with ODSs, and therefore nothing to do with 'recovery'. It results from CO2-induced cooling of the upper stratosphere. What then is 'super-recovery'?

Line 475: Ah, so the Montreal Protocol has effected a recovery in ozone from the effects of ODSs since the late 1990s? Your paper is communicating very mixed messages. Let me ask a very simple question: Is ozone in the lower stratosphere recovering from the effects of ODSs? If you answer yes, then what you have written elsewhere in the paper is wrong. If you answer no, then what you have written here is wrong because here you say that ozone declines would have been far worse without the Montreal Protocol which, to me, says that the Montreal Protocol has effected a recovery of ozone from the effects of ODSs. Please write clearly what you mean.

GRAMMAR AND TYPOGRAPHICAL ERRORS

Line 2: Replace 'work suggests' with 'work has suggested'.

Line 6: Replace 'wiped out' with 'offset'. 'wiped out' is too colloquial. Likewise on line 72.

Line 10: Replace 'hemispheric' with 'hemisphere'.

Line 18: Replace 'protocol's' with 'Protocol's'.

Line 30: Replace 'its amendments' with 'its amendments and adjustments' just to be complete.

Line 31: Replace 'coincided with' with 'led to'. They can't be coincident if they are separated by 11-13 years.

Line 60: Replace 'negative trends' with 'negative trends in ozone'.

Line 96: Replace 'This data was' with 'These data were'.

Line 98: Replace 'in context' with 'in the context'.

Line 105: Replace 'the averaging the two products' with 'the averaging of the two products'.

Line 128: Be consistent in the way you spell timeseries.

Line 204: I would suggest replacing 'increase' with 'increase in SH mid-latitude lower stratospheric ozone' just to be completely clear (or whatever region Chipperfield reported the change over).

Line 210: Replace 'as more data is added' with 'as more data are added'.

Line 232: Either 'the identification criteria were' or 'the identification criterion was'. The word criteria is plural.

Line 277: Replace 'in context of these' with 'in the context of these'.

Line 250: Replace 'Equatorial variability related' with 'Equatorial variability in ozone related'.

Line 251: Replace 'decreases' with 'decreases in ozone'.

Line 251: Replace 'to that of the' with 'to that at'.

Line 311: Replace 'Whilst' with 'While', unless you really do want to be very British.

Line 317: I would suggest replacing 'DLM trends estimated' with 'DLM trends in lower stratospheric ozone estimated' just to be totally clear.

Line 351: Replace 'large resurgence in 2017' with 'large resurgence in ozone in 2017'.

Line 357: Replace 'that the middle-stratosphere exhibits' with 'that ozone trends in the middle-stratosphere exhibit'.

Line 365: Replace 'have made' with 'has made'.

Line 387: Replace 'tropical' with 'tropics'.

Line 393: Replace 'indicated' with 'indicate'.

Line 396: Replace 'exclude, 50–60°' with 'exclude 50–60°,'

Line 412-414: This sentence needs a lot of help.

Line 420: delete 'extremely'.

Line 454: Replace 'spread on' with 'spread in'.

Line 467: Replace 'is likely' with 'are likely'.

---

## Author Comment (AC1) · 2 Aug 2019

The response is attached in a separate pdf.

Please also note the supplement to this comment:
https://www.atmos-chem-phys-discuss.net/acp-2019-243/acp-2019-243-AC1-supplement.pdf

---

## Author Response (AR1)

**Response to referee comments on "Stratospheric ozone trends for 1985–2018: sensitivity to recent large variability" by W. T. Ball et al**

**General comments relevant to both referees:**

We thank all referees (and the interactive comment of M. Diallo 16. March 2019) for their various important comments reproduced below, our replies are in blue text, while referee/reviewer comments are in black. Since almost all the points raised are specific points, rather than a generalized response, we respond to each one below point-by-point.

One change worth raising to the reviewers is that Fig. 4, showing annual mid-latitude and tropical changes in the QBO had a plotting error, and additionally, we changed the choice of altitude to 15 hPa as this better represents the message we are trying to make about the variability of ozone dependent on season and QBO phase. This also make the results consistent with Fig 2, with an upward ozone anomaly in the SH during a westerly phase QBO, the only extreme increase in that phase, that took place around the time of the only southern hemisphere sudden stratospheric warming in 2002.

**References needed**

Two paper by Diallo et al. (2018) and (2019) recently published in ACP discusses the combined influence of QBO and ENSO on the UTLS ozone and water vapour distributions using a lagged multiple regression analysis. A reference to these paper should be included along with a discussion of the added scientific value of the results presented here.

We thank M. Diallo for making us aware of these recent papers. These papers have now been references twice in relevant discussions of ENSO and QBO impact on both variability and ozone in the lower stratosphere (and in general throughout).

The papers are accessible here:

1. Diallo, M., Riese, M., Birner, T., Konopka, P., Müller, R., Hegglin, M. I., Santee, M. L., Baldwin, M., Legras, B., and Ploeger, F.: Response of stratospheric water vapor and ozone to the unusual timing of El Niño and the QBO disruption in 2015–2016, Atmos. Chem. Phys., 18, 13055-13073, https://doi.org/10.5194/acp-18-13055-2018, 2018.

2. Diallo, M., Konopka, P., Santee, M. L., Müller, R., Tao, M., Walker, K. A., Legras, B., Riese, M., Ern, M., and Ploeger, F.: Structural changes in the shallow and transition branch of the Brewer–Dobson circulation induced by El Niño, Atmos. Chem. Phys., 19, 425-446, https://doi.org/10.5194/acp-19-425-2019, 2019.

**Anonymous Referee #1**

This manuscript is an update and extension of a similar analysis by Ball et al. (ACP, 2018), which reported evidence from satellite data that ozone in the lower stratosphere at latitudes less than 60 degrees has continued to decline since 1998 even though upper stratospheric ozone has started to recover in response to the Montreal Protocol bans of

many ozone-destroying substances. Since much more ozone is in the lower stratosphere than in the upper stratosphere, the lower stratospheric decline dominates. On the other hand, evidence for an increase in tropospheric ozone was reported that could potentially cancel out the lower stratospheric decline. In that 2018 paper, no conclusion about the cause of the continued lower stratospheric decline was drawn and possible explanations involving both dynamics and chemistry were only briefly discussed.

In this manuscript, the trend analysis is slightly updated by the addition of more data and a stronger conclusion is drawn that the cause of the continued decline in the lower stratosphere is dynamical in nature rather than chemical. Specifically, the manuscript agrees with a recent study by Chipperfield et al. (GRL, 2018) that a strong positive ozone anomaly in 2017 was driven by short-term dynamical transport of ozone. However, it is further shown here that this short-term increase was caused by the stratospheric quasi-biennial oscillation and that a long-term gradual decline in lower stratospheric ozone remains even when the QBO-related variability is taken into account. The manuscript also agrees with the Chipperfield et al. conclusion that short-lived chemical depletion cannot explain the long-term decline, which must therefore be dynamical in origin.

I have two main comments on the manuscript, at least one of which can be considered as major. If the authors can satisfactorily address these comments in their revision, publication can be recommended.

(1) The magnitude of the long-term trends in zonal mean extrapolar lower stratospheric ozone is not clearly stated in the abstract or in most parts of the text. This leaves readers in the dark about how important the declines are. According to Figure 2a, the overall quasi-global decrease over 1985-2018 (33 years) is roughly 3.5 DU, which is in the range of 1.0 to 1.5%, or roughly -0.5% per decade. For comparison, the increase in tropospheric column ozone in the same latitude range estimated by Ball et al. (2018; their Figure 4) is +1.68 +/- 0.11 DU per decade or roughly +0.5% per decade. Could the authors please add numbers like this to the abstract and conclusions sections? The present manuscript notes large uncertainties in the actual tropospheric ozone increase (lines 38-41). However, at least the magnitude of the estimated lower stratospheric ozone decreases and the strong possibility of compensating tropospheric ozone increases should be more explicitly stated. Then readers can judge for themselves the importance of the observed lower stratospheric ozone decline due to dynamical processes.

We are happy to include some of these numbers in the main text, but rather avoid making the abstract less fluid and more difficult to read; these numbers are primarily included in the discussion and conclusions. In addition, we include an extensive table of numbers (absolute levels in 1985 and 1998, and DU, % and %/decade changes over 1998-2018, 1985-1998, and 1985-2018).

In addition, to be clear, the values mentioned above by the reviewer should relate to the quasi-global *lower* stratosphere. As such, the 1985-2018 (34 years) change is 3.4 DU, with a background level in 1985 of 87.8 DU (for the lower stratospheric component) leads to a change of -3.8% over the period, or approximately -1.1% per decade; this can be split into -1.3% per decade before 1998 and 0.9% after.

Finally, we do not wish to speculate on the tropospheric changes in the context of contribution to offsetting since we do not know how tropospheric ozone was changing prior

to 2005 (based on Ball et al., 2018), and other work indicating global tropospheric changes do not go back to 1985 (Gaudel et al., 2018).

(2) In this manuscript, as well as in the published work of Ball et al. (2018), the seasonal and longitude dependences of the observed zonal mean ozone trends are not evaluated. By not evaluating these characteristics or at least reviewing previous work on these characteristics, the authors are missing some important clues for understanding the origin of the observed zonal mean ozone trends in the lower stratosphere. Beginning in the 1990's, a number of authors have found that there is both a seasonal and a longitude dependence of column ozone trends, most of which originates in the lower stratosphere, especially at middle latitudes in winter and spring. Some of these authors concluded that these dependences are a consequence of decadal variability of quasi-stationary ultra-long Rossby waves that propagate from the troposphere into the stratosphere (Hood and Zaff, JGR, 1995; Peters and Entzian, Meteorol. Z., 1996; Peters et al., Beitr. Phys. Atmos., 1996; Hood et al., JGR, 1997; Hood et al., JGR, 1999). Decadal climate variability was therefore implicated. This was an extremely controversial and unpopular conclusion at the time because the prevailing view was that the observed ozone trends in both the polar and extrapolar regions in the lower stratosphere were dominantly chemical in origin. Note that these quasi-stationary waves are not necessarily linear (even though most mathematical treatments make this assumption) so that the zonal mean ozone change is not necessarily zero. Moreover, there is an associated trend in the BDC, which also affects ozone amounts, because the BDC is driven by Rossby wave breaking and absorption in the stratosphere (e.g., Hood and Soukharev, JAS, 2005). The continued decline of lower stratospheric ozone reported in this manuscript may therefore imply that there is a long-term component of the variability of these ultra-long waves. Schneidereit and Peters (Atmosphere, v. 9, p. 468, 2018) have recently investigated the zonally asymmetric component of ozone trends over 1979-2016 and find negative trends over Europe in winter that are twice as large as the zonal mean trend. A strong negative trend in one area such as this could produce a net change in the zonal mean ozone if the waves are non-linear. These authors suggest that long-term changes in the Rossby wave train that propagates out of the tropics and is linked to Arctic warming may be the ultimate driver. If the authors are not able to investigate zonally asymmetric ozone trends or long-term changes in quasi-stationary waves and their breaking behavior in the stratosphere for the present manuscript, then they should at least include some discussion of this previous work with references to some of the above papers.

Significant decreases in total ozone at (northern) mid-latitudes as published by the International Ozone Panel Report in 1988 were believed to be caused by anthropogenic emission of ODSs, which ultimately led to the Montreal Protocol and its enforcements. Subsequently, several studies were published presenting results that stratospheric ozone changes were (partially) attributed to long-term climate variability or other drivers causing additional trends (e.g. shown by Schneidereit and Peters, Atmosphere, v. 9,p. 468, 2018) indicating that ozone changes at particular latitude-longitude sites have large (winter) total ozone trends. Appenzeller et al. (2000), and Weiss et al. (2000) have discussed how winter ozone negative trends can be strongly enhanced by the NAO/AO over central Europe (i.e. Arosa). The EU project CATNDIDOZ also looked at similar questions regarding spatial influences on zonally averaged and global trends.

However, in this study, we have focused on zonally symmetric trends, which is of course a simplification of the real world. While of course the reviewer is correct that, in reality, the variability has a longitudinal component, and investigating this is worth undertaking, we do not make that step for now. Indeed, considering the non-zonal component might be taken

into account more properly by using equivalent latitude (at least at higher latitudes rather than the lower that dominate this study) instead of geometric latitudes averaged over some bands, and would be a natural way to account for the geographical, longitudinal inhomogeneity of ozone. For now this remains outside of the scope of this manuscript.

Nevertheless, it is worth mentioning at least in a brief discussion the potential physical drivers, particularly for future work. As such, we have added the following into the manuscript. 'Part of the negative trends in northern hemispheric stratospheric ozone in the 1980s and 1990s at higher latitudes have been previously attributed to synoptic and planetary waves (Hood & Zaff, 1995; Hood et al., 1999) inducing large localised (e.g. over Europe) wintertime decreases in ozone that might in turn be driven by sea surface temperature and eddy flux changes on decadal or longer timescales, although most of these studies are limited to the end of the last century when ODSs remained an established primary driver of the decrease. Nevertheless, these dynamical changes do not in themselves determine a specific underlying driving force, although increasing anthropogenic greenhouse gases (GHGs) is an obvious, though unverified candidate  (Hood & Soukharev, 2005; Peters & Entzian, 1999; Ball et al., 2018).

**Anonymous Referee #2**

GENERAL COMMENTS

This paper presents a compelling update to the earlier work of Ball et al., and a much awaited response to the Chipperfield paper. While the science is excellent, unfortunately the writing is often too curt and the reader has to guess at what is being referred to (see many examples below). The writing, in many places, indicates a lack of clarity of thought, or perhaps just that the author is not writing what he means to say. I would strongly recommend that the author improves the quality of the writing. There are no major changes required for this paper to be ready for publication. Most of my suggested changes below are relatively minor. It is an excellent piece of analysis. Now it just needs to be communicated to its audience clearly.

We appreciate the author's positive comments. Additionally we thank the reviewer for the clear amount of time and effort that he/she has made to ensure that we do not miscommunicate our results, nor the success of the Montreal Protocol and its amendments.

SPECIFIC COMMENTS

Line 1: I would suggest replacing 'The Montreal Protocol' with 'The Montreal Protocol, together with its amendments and adjustments'.
Replaced with 'The Montreal Protocol, and its subsequent amendments, …'

Line 3: I would suggest that you reserve the word 'recovery' strictly to talk about the recovery of ozone from the effects of ODSs. When you are talking more generally about ozone increases, whether driven chemically or dynamically, talk rather about 'ozone increases' rather than 'ozone recovery'.

We agree that it is important to be clear on this issue. Following this, a similar suggestion by reviewer 3, and multiple specific suggestions regarding this point later in this review, we

have systematically gone through the manuscript to replace 'recovery' with 'increase'/'decrease' or to specify that the recovery has to do with 'ODSs' unless it is already clear.

This however raises an interesting point regarding terminology because, like the reviewer, we expect 'recovery from ODSs' to be already occurring everywhere due to the reduction of chlorine in the atmosphere, and yet we do not see signs of it in the lower stratosphere. If we assume that to be the case, then recovery is underway everywhere. This then suggests the need for new terminology in light of counteracting influence of GHGs. For example, perhaps 'restoration' to pre-ODS ozone-levels is a term that should be adopted to represent 'a return to levels prior to anthropogenic influences'. For now, we will leave this definition aside in the manuscript.

Line 12: I am struggling a bit with this 'still likely lower than in 1998 (probability ~80%)'. Surely it is either lower or it isn't. OK , maybe I will get to see later how this is nuanced by some statistical significance.
We do not use frequentist significance tests in this paper, but rather make straightforwardly interpretable Bayesian probability statements that reflect "the probability (or confidence) that X is true given the observed data, model and prior assumptions". In the case you highlight for example, we calculate the (Bayesian posterior) probability distribution for the overall change since 1998 given the observed data and model assumptions, and estimate from this the fraction of the PDF that is either negative or positive. In this case, the overall change since 1998 is likely negative with 80% probability, and 20% probability that it is positive ( because those are the fractions of the probability distribution that are split by the zero line). Our use of Bayesian probabilities is intentional  and makes for easily interpretable and robust statements. Frequentist significance statements are muddied by the fact that they are not "direct" inferential statements, so misinterpretation is easy (and commonplace), and too often lead to the poor practice of drawing binary inferential conclusions (X is proven/rejected) based on arbitrary thresholds (eg p < 0.05).

Lines 16-17: With regard to the sentence 'These decreases do not reveal an inefficacy of the Montreal Protocol', an important point that can, and perhaps should, be made here is that tropical stratospheric ozone is almost certainly recovering (from the effects of ODS), and hence the Montreal Protocol is successful, while simultaneously still declining (due to other influences). The point to make is that recovery and declining ozone are not mutually exclusive.
We agree. However, since we do not diagnose either way the ODS-recovery and GHG-related-decline, nor attribute the ratio of these factors, we do not wish to speculate or assume that point; such quantification should be (and has been) performed using chemistry climate models. Since we make the point in the following sentence, that the Montreal Protocol has been beneficial, and that dynamics are probably in play in counteracting it, we do not think this needs adjusting.

Line 33: You need to be clear here what you mean by 'significant'. Do you mean statistically significantly different from zero at the 2 sigma level, or do you mean a more general 'significant' as in 'large'. It matters a lot in this specific context so I think that you should be clear.
Agreed, and this point was raised by another reviewer. We have added 'statistically' before 'significant'.

line 59: I think it would be plausible to say that climate change may be exacerbating some specific dynamical mechanism (or more than one) that affects ozone, but it feels incongruous to state that climate change, in and of itself, could be a mechanism for dynamically affecting ozone.

Agreed, and this point was raised by another reviewer. We have replaced 'climate change' with 'increasing anthropogenic greenhouse gases (GHGs)'.

Line 125: Delete 'point the reader to Laine et al. (2014) for details on this method and' since you have anyway cited the Laine et al., 2014 paper.

Done.

Line 135: This essentially assumes that the sensitivity of ozone to the regressors is time-dependent. What support is there for this assumption? Why might ozone have a certain sensitivity to EESC in 1985 and then a different sensitivity in 1995, or 2005? This ability for DLM to accommodate changes in the sensitivity of ozone to the regressors is presented as an advantage of DLM over MLR but I am not convinced that it is. In choosing DLM over MLR you are making some significant assumptions and I am not sure that you have support for those assumptions. Or is it just the amplitude and phase of the seasonal cycle that you allow to vary with time?

Our main reason for supporting DLM over MLR is its superiority over MLR in attaining a better estimate for trends in validation-test cases where the trend is known in advance (Ball et al., 2017) and because, after that validation, DLM provides more insight into the evolution of the trend than MLR does (when assuming linear or piecewise-linear trends). You are correct: in the DLM model used for this paper, only the amplitude and phase of the seasonal components are allowed to vary in time, not the other regressor coefficients. We note also that since, in ozone composites, the instruments change in time, so do the observing kernels and the exact region observed; this can introduce an artificial apparent change in amplitude of the seasonal cycle and motivates allowing for variation of seasonal (and regressor) parameters in time. We allow this because the seasonal part has the largest variance and if the seasonal cycle changes with time then removing a mean seasonal cycle may lead to a bias in the residuals over time; we see this also as an advantage. It may well be true for the other drivers, but this does not have a strong justification yet. It should be noted that the seasonal cycle, even when evolving, is required to be stationary over time to avoid leaking into the non-stationary trend component. Many of the issues related to MLR are also discussed in Petropavlovskikh et al., 2018 (SPARC Report).

Line 138-139: Well, unless EESC is selected as a descriptor of the long-term secular trend in which case it is a much more natural and appropriate descriptor than any pure statistical descriptor.

We agree that EESC is a key component of the long-term trend and it is hence natural to include it as a regressor, but even then it is an overly strong assumption to assert (in the model) that the only thing driving the background trend is EESC. We also note that the EESC will need a second component to allow for spatial variation in the inflection point, otherwise the same issue with respect to the piece-wise linear misrepresenting changes will occur. A GHG regressor would also make sense as an additional component of the background trend. However, here we simply aim to diagnose the long-term trend, whether ODS or GHG (or other) related, following the standard aim in this community. The DLM trend has the advantage that it does not tie the background trend to a particular prescription for the shape or a (possibly incomplete) set of components before the data are analyzed; it is agnostic about the shape or drivers of the trend. However, for the same reason, the DLM trend term does not elicit specific physical drivers of the background trend on its own. We

have added in bold/italic: 'Secondly, MLR that does not assume a driver for the long-term trends, e.g. for the influence of ODSs or GHGs typically assumes a fixed prescription for the shape'

Lines 143-144: I don't be believe it is true that 'in practice MLR is often performed by first subtracting an estimated mean seasonal cycle'. I certainly don't. Most MLR-based analyses I have seen fit the annual cycle as a series of Fourier expansions along with all of the other regressors.
We agree that not all analyses subtract the seasonal cycle, e.g.: Kyrola et al., 2013 (ACP); Frith et al., 2017 (ACP); Petropavlovskikh et al., 2018 (SPARC). But many do and this was how, e.g.: Maycock et al., 2016, 2018 (ACP); Sofieva et al., 2017 (ACP); Steinbrecht et al., 2017 (ACP). We have added in bold/italic: '…and then making a post-hoc correction for auto-regressive residuals, *although many do fit annual and semi-annual components*.'

Line 151: What is MCMC? I have not seen this acronym defined anywhere.
Agreed – we have added 'Monte Carlo Markov Chain' prior to the first mention of MCMC'.

Line 170: Can you give some indication of why the SAOD time series is not available beyond 2016?
The dataset had not been extended at the time of the manuscript. Thus, we have modified the sentence to read: '…for this analysis, the SAOD is *currently* not  *extended* beyond 2016, so we repeat the year 2016…'

Line 196: It wasn't clear to me what was meant by 'this group of spatial responses'. Responses of what to what? Are you referring the latitude/pressure resolved trends plotted in Figure 1?
Yes. We have modified this sentence to read: 'any sensitivity of the end year to the state of these drivers should be encapsulated in  *the set* of spatial responses *depending on the end year only (Fig. 1)*…'

Line 214: But the latitudinal extent of the changes you are seeing in observations is much wider than what is seen in the CCMs right?
Yes, though we cannot attribute that, with trend analysis only, to the BDC, which is why simply leave this as a statement, and only with respect to the MMM of CCMs (see WMO 2014 and 2018 reports), not to individual members of the MMM.

Line 240: But only in the lower stratosphere right?
An excellent point; we have modified the sentence: '…convincingly showed that the majority of post-1997 quasi-global *lower stratospheric* ozone variability…'

Line 253: I think that you need to read and cite Gray, L.J. and Pyle, J.A., A twodimensional model of the quasi-biennial oscillation of ozone, J. Atmos. Sci., 46, 203-220, 1989. Another paper that might be relevant is Bodeker, G.E.; Garny, H.; Smale, D.; Dameris, M. and Deckert, R., The 1985 Southern Hemisphere mid-latitude total column ozone anomaly, Atmos. Chem. Phys., 7, 5625–5637, 2007, especially if you are seeking clarification of the origin of the large mid-latitude changes in ozone that occur every few years.
We thank the reviewer for these suggestions, which we have integrated into this paragraph, including the addition of: '*Bodeker et al. (2007) previously identified large SH negative anomalies in 1985, 1997 and 2006 and related these to the QBO-Westerly phase}*'. We note that it appears to us that the general relation between QBO and ozone is rather well known in the science community, although the connection of the event highlighted by Chipperfield

et al. (2018) and the additional ones of similar magnitude we have identified, to the QBO has not yet been made until here. We further believe this motivates the need to improve attribution methods to account for these interactions.

Line 311: It is not clear to me what you mean by 'governing each other'? Do you just mean 'governing each'?
You are correct, we do, so we have clarified this: '…the processes governing each  *timescale* are likely quite different.'

Line 325: This is worded in a very confusing way - please rewrite.
We have rewritten this as follows: 'This agrees with Chipperfield et al., 2018 who suggested the large  *rapid increase* of 2017  affected trends, although this was mainly in the SH *and has subsequently showed little change over 2018*.'

Line 329: As *what* in years prior to 2010 are essentially unaffected by the addition of 2017 and 2018? Ozone in 2013 is unaffected by the addition of 2017 and 2018? You are referring to something being unaffected by the addition of 2017 and 2018 but I am not sure what that something is.
We agree this is confusing. The 'what' is '*the DLM-estimated change of ozone relative to 1998 in*'; this has been inserted at the start of the sentence.

Line 334: You need to make it clear that you are referring to the minimum in the DLM fit and not a minimum in the observed ozone.
We have clarified this: 'If that happens it is therefore possible that the non-linear trend estimates will likely decrease again,  *and* the emergent 2013 minimum *in the DLM non-linear trend estimate* seen in Fig. 2b *is* likely to shift to a later date or disappear.'

Line 336: Do you mean mid-latitude ozone excursions depend on the phasing of the QBO phase from westerly to easterly (or vice versa) on the phase of the annual cycle in ozone? If so, please consider wording as such. If not, please reword to be more clear.
We have added in bold/italic: 'Since mid-latitude ozone excursions depend on the QBO-seasonal interaction*, i.e. the QBO phase relative to the time of year,* …'

Line 337: Can't you just have cross-terms in your regression model i.e. a QBO basis function modulated by a phase dependent seasonal cycle?
You could do that, but the phase of both QBO and seasonal terms will depend on location, and simply multiply terms does not work. As the phase of the two terms should not simply be estimated by a minimisation of, e.g., the residuals, but should have some underlying physical motivation connected to it, we do not do it here and leave it for future work.

Line 338: I don't think that is true. They can, they just need to include the appropriate regression model basis functions. Perhaps many MLR models currently in use do not, but that does mean that they fundamentally can't.
Absolutely. It is true given these predictors don't use it; but there is no standard way to deal with it yet, so it remains true until this has been resolved (see previous point above) and why we have stated the sentence with 'without predictors…'.

Line 358: Uncertainties in what are consistently large?
We agree this is unclear, and have added in bold/italic: '…exhibits the largest sensitivity to the end year and uncertainties *in the change from 1998* are consistently…'

Line 359-360: The upper stratosphere is also sensitive to what in the tropics? Too often the explicit subject of a sentence or phrase is omitted in your writing which requires the reader to constantly be guessing at what you are referring to. This makes deciphering the narrative very tiring. There are many examples of this in my comments. The next one is on the same line (360):

We have clarified this as: 'The upper stratosphere is also sensitive **to the end year** in the tropics'.

Line 360: What has 'shifted from negative to positive'. And what, exactly is uncertain?

We have clarified this too: '…  **and the end year**  shift **the estimated ozone change** from negative to positive **with ncreasing end year**, although **the uncertainty** always  **remains large.'**

Line 361: Uncertainties in what are smaller? And smaller compared to what?

We have clarified as: 'at mid-latitudes uncertainties **in the change since 1998** are smaller…'

Line 361: There has been a general shift in what towards more positive and significant increases? Please read the sentence that starts on line 359 and ends on line 362 i.e. "The upper stratosphere is also sensitive....SH and quasi-global estimates" and see whether, read as it stands, it would make sense to someone. Take that sentence to a colleague and ask them to tell you what it means. They may be horrified to read that the stratosphere has shifted from negative to positive. Maybe they always thought that the stratosphere was positive and may be alarmed that it has become negative. But at least they will know that whatever happened, its (whatever it is) is always uncertain. I would strongly suggest that you write in a way that prevents the reader from having to guess at things.

This is a fair criticism. We have amended the text to be clearer following the reviewer's suggestions.

Line 367: The statement that 'The quasi-global lower stratosphere continues to exhibit a monotonic decline' is not true. There are many things in the quasi-global lower stratosphere that are not continuing to exhibit a monotonic decline. A good example would be CO2. Please work to improve the precision of your writing.

Agreed – we have inserted 'ozone' into this sentence to clarify it is only ozone we are assessing.

Line 368: I was shocked to read that 'the whole stratosphere continues to remain lower than in 1998'! Is the sky really falling? https://www.youtube.com/watch?v=NO04VXBIS0M. Is there nothing we can do to lift the stratosphere?

We did not mean to shock the reviewer, nor future readers. As such, we have amended the sentence as: '…and **ozone abundances integrated over** the whole stratosphere **continues to** remain lower …'

Line 374: Regarding 'changes prior to the last five years are largely unaffected in the partial columns'. I would be horrified if it was possible that ozone prior to the last five years was affected by the addition of recent years. It would mean that someone, somewhere, has invented time travel. But perhaps that's not what you mean? It might be a good idea then to write *exactly* what you mean.

Again, the reviewer makes a fair point. We have modified the text here as follows. Figure 5 also confirms that the **gradients of the** non-linear curves are only affected by unmodelled variance in years close to the end points, **typically within the last five years of the partial**

*column timeseries considered here; the shape of the DLM curves*  prior to the *final*  five years *of the DLM curves* are largely unaffected .'

Line 403: So are you really saying that the Montreal Protocol is working only in the upper stratosphere and not in the lower stratosphere. This will hugely concern policymakers. They will wonder why all the hard work they have done since 1987 in reducing emissions of CFCs, halons, HCFCs and other ODSs has only decreased their concentrations in the upper stratosphere. Could I put it to you that the Montreal Protocol has been effective in reducing ODS concentrations, and thereby concentrations of Cly and Bry throughout the atmosphere, and that, as a result, ozone throughout the atmosphere, including the lower stratosphere, is recovering from the effects of those ODSs. Is this recovery apparent in observations in the upper stratosphere? Apparently yes. I say apparently only in that (at least in this paper) a thorough attribution of the drivers of those ozone increases has not been done. Is this recovery apparent in observations in the lower stratosphere? No, clearly not? Why not? Well because other factors have been affecting ozone (not diagnosed in this paper) that are likely (we cannot be sure since a thorough attribution has not been done) overwhelming the increases brought about by reductions in concentrations of Cly and Bry. Wouldn't that be a more accurate picture to communicate to policy-makers?

In order to be precise, we have included the following at line 403:

*'An upper stratospheric increase is the expected result from long-term stratospheric chlorine reductions, a direct consequence of the Montreal Protocol and its amendments, though we do not explicitly attribute the cause of the increase to that here. Indeed, the Montreal Protocol and its amendments will have been effective in reducing ozone losses throughout the stratosphere through reductions in CFC emissions, HCFCs and other ODSs. The lack of a positive trend since 1998 in the lower stratosphere, as opposed to the one clear in the upper stratosphere, is likely the consequence of other factors such as dynamical changes (Wargan et al., 2018).*'

Line 412: What does it mean to be 'confident in the evolution of stratospheric ozone'? You're confident that ozone is evolving? I'm pretty confident it is too. It always has been. I don't need your paper for that. Maybe you mean that the aim of this work is to build confidence in our quantitative understanding of trends, and other long-term variability, in upper, middle and lower stratospheric ozone?

Agreed. We have replaced this sentence with: '*The aim of this work is to assess the current state of, and trends in, stratospheric ozone. Improved knowledge of such trends, and the relevant forcing mechanisms and associated variability, will help to better constrain CCM projections of ozone to the end of the 21st Century.*'

Line 414: I think that the best tools for studying long-term changes in ozone, and attributing the causes of those changes to known drivers, is the application of regression models to observations. Yes, models can be useful for attribution but they have little role, if any, in detecting the changes in the first place. I think that chemistry-climate models are the best tools for making projections of how ozone may change in the future.

We certainly agree that they do not have as strong a role in initial detections, especially in this case where the implication is models are not reproducing these trends. However, we still believe these remain an excellent tool. So we have inserted 'one of' in this sentence: 'Chemistry models resolving the stratosphere are *one of* the best tools for attribution and long-range studies of ozone'.

Lines 423-425: This sentence blurs the lines between observations and model output. Were the 2017 data from Chipperfield CTM output or observations? If it was model output I would

suggest replacing 'found lower stratospheric ozone had rapidly increased in 2017 back to 1998 levels' with 'found that model simulated lower stratospheric ozone increased in 2017 back to 1998 levels'.

As also mentioned by another reviewer, we have modified this sentence partly following your advice. 'A recent study (Chipperfield et al., 2018) used a CTM to reconstruct  the ozone timeseries *beyond the observational record available at the time to 2017 and found that that model simulated a lower stratospheric ozone increase in 2017 back to 1998 levels*; this was attributed to dynamical variability.'

Line 429: Any idea why the CTM got it so wrong?

No, we do not know why.

Line 438: Does it enhance the positive trend in ozone, or does it enhance the recovery of ozone from ODSs - noting that those are two very different things?

Fair. Modified to reflect the former: 'We also find that the 2017--2018 addition enhances the *estimated* magnitude of the upper stratospheric ozone  *positive trend*'

Lines 439-440: How can the 'recovery' display a 'reduction'. That makes no sense to me. I can understand how ozone can reduce. I can even understand that ozone can reduce while simultaneously recovering from the effects of ODSs (I am not saying that that's what is happening here, but it is conceivable).

A reread is also partially confusing to us too. We have amended it, striking out 'recovery of the' to simply state that it still displays a reduction.

Line 447: By 'continues at all latitudes north of 30°S' do you mean continues to decrease at all latitudes north of 30°S?

Thank you for this clarification; 'to decrease' has been inserted.

Line 449: The seasonal-dependence of the QBO on what? Or do you mean the seasonal dependence of ozone on the QBO?

Yes; inserted 'ozone on'.

Line 455: This is the first mention of 'return dates'. What is meant by this? What is the 'return date' in a CCM?

Yes, this need clarifying: '…and therefore their return dates, **i.e. a return of ozone to the level it was in 1980 (WMO 2014, Dhomse et al., 2018, WMO 2018)**.'

Lines 456-457: What, exactly, do you mean by 'numerical inaccuracies'? Can you please add a sentence or two that elucidates this.

We meant 'numerical diffusion', which has replaced 'inaccuracies'.

Line 460-461: Wait a minute. I have seen no evidence anywhere that there has been a 'halt in the recovery in total column ozone' from the effects of ODSs. I have seen plenty of evidence that ozone in different regions of the atmosphere continues to decline (including this paper) but no attribution of this declines such that one could conclude that ozone is not recovering from the effects of ODSs. In fact it would devastate our understanding of atmospheric chemistry if it was found that decreasing concentrations of Cly and Bry had no impact whatsoever on the Cly and Bry cycles that destroy ozone. I see chemists leaping from buildings. So I strongly reject your conclusion that ozone, anywhere in the atmosphere, is not recovering from the effects of ODSs as you have presented no evidence at all to that effect. To make that call, you would need to do a robust attribution of changes in ozone to

date and demonstrate that the ozone changes attributed exclusively to changes in Cly and Bry have been negative. And that you have not done.

A fair point. We have clarified this as follows: 'The halt **in ODS-related ozone losses as a result of the Montreal Protocol and its amendments,** and **an** initial recovery in total column ozone is almost universally reproduced by CCMs…'

Line 464: Do you mean predictions or projections? I think that it is very dangerous to use models to make predictions.

We meant 'projections' and have swapped these words accordingly.

Line 468: Do you mean total column ozone or do you mean ozone in all parts of the atmosphere?

We meant total column, and have clarified this.

Line 470: I have no idea what you mean by 'super-recovery'. I fully understand how ozone can recover from the effects of ODSs. But I can't understand how it can 'superrecover'. I can understand how ozone could become higher than it was in the 1960s, but this has nothing to do with ODSs, and therefore nothing to do with 'recovery'. It results from CO2-induced cooling of the upper stratosphere. What then is 'superrecovery'?

Super-recovery is a term used by the WMO ozone assessment report (and other articles); from the WMO 2018 ozone assessment: *'For global mean total ozone columns, the return to 1980 values is faster and the possibility of super-recovery (i.e., the increase of ozone above historical levels) is higher for the RCPs with larger GHG increases.'* We have also modified the error of a return to '1960s' levels to '1980s'.' We have added a similar clause as the WMO to our manuscript: '…continuing on to a `super-recovery', **i.e. that ozone will be higher by the end of the 21$^{st}$ Century than prior to 1980s levels**…'

Line 475: Ah, so the Montreal Protocol has effected a recovery in ozone from the effects of ODSs since the late 1990s? Your paper is communicating very mixed messages. Let me ask a very simple question: Is ozone in the lower stratosphere recovering from the effects of ODSs? If you answer yes, then what you have written elsewhere in the paper is wrong. If you answer no, then what you have written here is wrong because here you say that ozone declines would have been far worse without the Montreal Protocol which, to me, says that the Montreal Protocol has effected a recovery of ozone from the effects of ODSs. Please write clearly what you mean.

Agreed. Thanks to the time you have invested, and detailed comments, this manuscript should now be very clear and consistent by this point. We appreciate that.

GRAMMAR AND TYPOGRAPHICAL ERRORS

Line 2: Replace 'work suggests' with 'work has suggested'. Done.
Line 6: Replace 'wiped out' with 'offset'. 'wiped out' is too colloquial. Likewise on line 72. Done.
Line 10: Replace 'hemispheric' with 'hemisphere'. Done.
Line 18: Replace 'protocol's' with 'Protocol's'. Done.
Line 30: Replace 'its amendments' with 'its amendments and adjustments' just to be complete. Done.
Line 31: Replace 'coincided with' with 'led to'. They can't be coincident if they are separated by 11-13 years. Done.
Line 60: Replace 'negative trends' with 'negative trends in ozone'. Done.
Line 96: Replace 'This data was' with 'These data were'. Done.

Line 98: Replace 'in context' with 'in the context'. Done.
Line 105: Replace 'the averaging the two products' with 'the averaging of the two products'. Done.
Line 128: Be consistent in the way you spell timeseries. Done.
Line 204: I would suggest replacing 'increase' with 'increase in SH mid-latitude lower stratospheric ozone' just to be completely clear (or whatever region Chipperfield reported the change over). Done.
Line 210: Replace 'as more data is added' with 'as more data are added'. Done.
Line 232: Either 'the identification criteria were' or 'the identification criterion was'. The word criteria is plural. Done.
Line 277: Replace 'in context of these' with 'in the context of these'. Done.
Line 250: Replace 'Equatorial variability related' with 'Equatorial variability in ozone related'. Done.
Line 251: Replace 'decreases' with 'decreases in ozone'. Done.
Line 251: Replace 'to that of the' with 'to that at'. Done.
Line 311: Replace 'Whilst' with 'While', unless you really do want to be very British. The lead author is very British; but we have made the amendment.
Line 317: I would suggest replacing 'DLM trends estimated' with 'DLM trends in lower stratospheric ozone estimated' just to be totally clear. Done.
Line 351: Replace 'large resurgence in 2017' with 'large resurgence in ozone in 2017'. Done.
Line 357: Replace 'that the middle-stratosphere exhibits' with 'that ozone trends in the middle-stratosphere exhibit'. Done.
Line 365: Replace 'have made' with 'has made'. Done.
Line 387: Replace 'tropical' with 'tropics'. Done.
Line 393: Replace 'indicated' with 'indicate'. Done.
Line 396: Replace 'exclude, 50–60◦' with 'exclude 50–60◦,' Done.
Line 412-414: This sentence needs a lot of help. A lot of help has been provided to this sentence; done.
Line 420: delete 'extremely'. Done.
Line 454: Replace 'spread on' with 'spread in'. Done.
Line 467: Replace 'is likely' with 'are likely'. Done.

**Anonymous Referee #3**

Ball et al. provide an update on Ball et al., further examining trends in stratospheric ozone derived from satellite observations. Additionally, they discuss the recent results of Chipperfield et al. (2018), who modelled large increases in lower stratospheric ozone in a CTM, indicating that the observations used in this study give a smaller increase and suggest that this is consistent with interannual variability driven by the QBO. Further, they suggest that, if this is the case, then the lower stratospheric ozone increases will decrease in the near future. Despite these increases, longterm trends in lower stratospheric ozone trends remain negative. The analysis and discussion presented in the paper is of a high standard and explores an important and relevant topic within the scope of SCP, and as such merits publication following revision. I have several comments the authors should address before publication:

General Comments:

1. As a major point of consideration, the authors frequently use the term recovery, or lack of, when discussing ozone increases and decreases. However, recovery of stratospheric ozone is really reserved for increases of ozone resulting from reductions in stratospheric Cly expected due to the Montreal Protocol. Changes resulting from dynamical variability or stratospheric cooling resulting from CO2 increases are not strictly recovery. As the manuscript does not explore all the drivers of ozone changes at different latitudes/altitudes, I feel the authors should refer only to ozone increases/decreases where they are not attributing ozone changes to change in ODS.

We accept this point. Repeating part of our response to reviewer 2's specific comment (regarding Line 3): we agree that it is important to be clear on this issue. Following this, a similar suggestion by reviewer 3, and multiple specific suggestions regarding this point later in this review, we have systematically gone through the manuscript to replace 'recovery' with 'increase'/'decrease' or to specific the recovery is to do with 'ODSs' unless it is already clear.

2. A second major comment is that the authors should state specifically what they are discussing in each section. Too often terms like decline/recovery/trend are used with no reference to ozone. This is particularly true in the sentence on L367-369, which states 'The quasi-global lower stratosphere continues to exhibit a monotonic decline that is still highly confident with 99% probability (Fig. 7 and Table 1), and the whole stratosphere continues to remain lower than in 1998. . .'. Further, more clarity should be provided so the reader knows when the authors are discussing ozone trends and ozone values.

The reviewer raises an important point, and we have made an effort to improve clarity throughout. Without writing an exhaustive list, the two examples were adjusted as such (bold/italic text):
- 'The quasi-global lower stratospher*ic ozone* continues…'
- '…and *ozone integrated over* the whole stratosphere continues to remain lower than in 1998…'

3. There is no discussion in the manuscript on the chemical lifetime of ozone – the reason that dynamical variability plays such a key role in the lower stratosphere is that here the chemical lifetime is long. A brief description of this fact, with references, would add to the introduction.

We have added a brief discussion of this when discussing the CTM in the introduction: "*Indeed, chemistry and photochemistry play a dominant role over dynamical perturbations in the upper stratosphere as ozone lifetimes are short (~days), while ozone lifetimes of ~6-12 months in the lower stratosphere means that equator-to-mid-latitude transport of similar timescales plays an important (dominant) role there (London, 1980; Perliski & London, 1989; Brasseure & Solomon, 2005).*"

4. The authors frequently refer to 'climate change' as a potential driver of the examined dynamical variability. However, I feel they would be better served by using a phase such as 'changes to stratospheric dynamics resulting from anthropogenic greenhouse gases,' which is closer to a mechanistic analysis. Climate change is itself a response to changing GHGs.

We agree that formally we are inaccurate using the 'climate change' as a forcing term; as such we have modified the following:

- 'Although the underlying driving force has not yet been determined,  *increasing anthropogenic greenhouse gases (GHGs)* is an obvious though unverified candidate.'
- '…rather than  *a response to increasing GHGs*…'
- '…due to enhanced upwelling from the Brewer Dobson circulation (BDC) as a result of  *changes to stratospheric dynamics from increasing GHGs*…'
- '…it remains to be seen if this can be attributed to the  *anthropogenic GHG* induced upwelling of the BDC…'
- '…this is important given that the changing climate *due to anthropogenic GHG emissions* may impact inter-annual dynamical variability…"

5. I miss any discussion on why the CTM results of Chipperfield et al. and the observations presented in this study differ so greatly. Is there any consensus on why this is? Does it represent some failing of the chemistry in the CTM? Is the re-analysis dataset used not accurate? IS it within the uncertainties of the observations? Further discussion on this point would improve the manuscript.

The reviewer offers a series of excellent questions worthy of an answer. However, there is no answer now, and we believe this would require exploring the model, as well as developing improved uncertainty estimates. We do not have explicit uncertainties on the observations, though MLS usually shows high precision and is stable, so these are expected to be smaller than the difference over 2017 between the CTM and observations. The CTM shows periods of similar deviations with respect to the observations (see 2003, 2007-2008 in Fig 1c of Chipperfield et al., 2018 for the lower stratosphere, although in that plot there is not data from the observations to directly compare). It may be that the issues relate to the CTM parameters or to the reanalysis fields driving the dynamics; but this is not something we wish to speculate on, but should indeed be investigated. We have simply added to the end of the second paragraph in section 3.2 discussing the difference between observations and CTM: '*We do not know why the CTM and observations disagree in the magnitude of change for this period.*'

Specific comments:

L1: Add 'and its subsequent amendments' following Montreal Protocol
Done.

L6: Replace 'wiped out'
Replaced with 'offset'.

L18: Important here to say that dynamical variability is counteracting the effects of the Montreal Protocol on stratospheric ozone recovery, not on the regulation of halogenated ODS.
Agreed. Sentence now reads: 'Rather, they suggest other effects to be at work, mainly dynamical variability on long or short timescales, counteracting the *positive effects of the Montreal Protocol on stratospheric ozone recovery* '

L19: Swap '(30-60)' with 'variations'
Done.

L33: Add 'statistically' before significant to indicate that you do not mean large increase

Done.

L33-43: The authors could cite here, or elsewhere in the manuscript, that these observations are supported by CCM studies (e.g. Meul et al., 2016; Keeble et al., 2017; Dhomse et al., 2018).

The suggested papers have varying degrees of conclusions regarding significant trends, the projection scenario and models used, and in what they specifically report, so we do not wish to overly complicate the introduction given the focus on observations only. However, we agree with the reviewer that this is worth pointing out, and so we have included the sentence: *'The importance of considering tropospheric and stratospheric changes separately to understand changes in total column ozone has also been highlighted in recent studies using chemistry climate models (CCMs) (Meul et al., (2016), Keeble et al. 2017, Dhomse et al., 2018}.'*

L44-49: Stress that these composites are observations
Done: 'To assess trends in stratospheric ozone, composites *of observations* must be formed by merging multiple ozone *observational* timeseries into a long, multi-decadal record…'

L45: Use consistent spelling of timeseries throughout the manuscript
Done; replaced all instances of 'time-series' with 'timeseries'.

L60: Add 'ozone' between 'negative trends'
Done.

L93: Remove 'see'
Done.

L94: IS the Froidevaux et al. paper cited her now published? If not I would recommend removing it from the manuscript, instead saying that the v2.20 dataset used here is an update of Froidevaux et al. (2015).
Froidevaux et al, 2019 is now publish; we have updated the reference.

L125: delete 'point the reader to Laine et al. (2014) for details on this method and' – the paper is already cited in the sentence
Done.

L135-137: Why do the authors allow the seasonal component to vary but keep the regressors constant? Is it more likely that one should vary in time than the others? Some additional text explaining the rationale behind this decision is warranted.
We have added:
The reviewer raises a good point. Because the seasonal cycle has the largest variance by far, and because there is some evidence it modulates with time either due to different instrument sampling or real changes in the seasonal cycle, we allowed it to freely vary. However, we accept that this could also influence other regressors and in future these should be allowed to vary too, though we leave an adoption of that freedom to a future assessment of the impact on trend analysis from allowing such freedom. We have modified this part of the text to include the bold/italic in the following. "Here, we allow the amplitude and phase of the seasonal components to be dynamic, but keep the regressor amplitudes constant in time; *we do this because the seasonal cycle in the observational composites can change over time either as a physical feedback of changing temperature and ozone, or*

*due to different observations exhibiting different seasonal amplitudes (not shown) that
are a result of the observing instruments `seeing' slightly different parts of the atmosphere
or having different sampling. Due to the seasonal cycle having the largest variability of all
modes we expect that, if left unaccounted for, the time varying seasonal modulation might
have an influence other parts of the regression. In principle other regressor amplitudes
could also have some time modulation for similar reasons.* We leave an *investigation of
more flexible DLM models with dynamic regressor amplitudes* to future work *where a
physically-motivated justification for such freedom can be investigated.*".

L137-140: This is not true if EESC is used to represent the long-term trend.
We agree in part, however, we would argue it is more complicated and our assertion
remains reasonable. The long term 'background' component, that which the DLM non-linear
trend or a piece-wise linear trend is supposed to represent, is likely to represent the impact
of GHG and ODSs (or EESC). Due to the need to account for a lag required in an EESC
regressor, a second EESC curve is needed to represent a shift in the EESC; either way these
are fixed shapes. The DLM does not require a fixed background trend prescription.
Therefore, we would argue that our description is reasonable. Additionally, we would argue
it is true that a piecewise linear trend does not represent the temporal evolution of EESCs,
GHGs, and their combined effects, especially when the inflection date is fixed the same for
all altitudes and latitudes.

L128-155: Are there any references in the literature to support the claims made in this
section? There is a lot of literature on using MLR techniques when determining ozone
trends, and any literature which assesses the DLM technique should also be cited.
It is true there is limited literature on this. Laine et al., 2014 and Ball et al., 2017 are the
main ones for ozone, with the former comparing results DLM with MLR on real timeseries,
and the latter considering idealized (simulated) test cases to compare performance. Ball et
al., 2017 showed that, in the validation-test cases they considered (where the background
trend is "known"), DLM out-performed MLR on trend reconstruction. A comprehensive
inter-comparison of the two approaches is still needed, but the description as laid out here
is formally correct, regarding the propagation of errors etc, so still provides a sound
foundation for motivating DLM over MLR. We have cited these two papers again at an
appropriate point in this section.

L151: What is MCMC – it does not appear to be defined in the manuscript.
MCMC means Monte Carlo Markov Chain, a type of algorithm for sampling probability
distributions; this has been added to the manuscript.

L197-202: The authors should be clear here that they are discussing ozone trends
This has been clarified.

L204: Add 'in lower stratospheric ozone' between increase and reported
Done.

L225: Replace 'upswing' – also elsewhere in the manuscript.
Done.

L240-241: specifically in the lower stratosphere, where the lifetime of ozone is long – please
clarify that in this sentence.
Added the part sentence as the reviewer suggested here.

L250: The authors could add here further discussion, with appropriate references, discussing other drivers of variability.
We have added the following:
'…but the variability is less regular, unsurprisingly since the NH stratosphere is known to have additional variability *, a consequence of greater sea-land contrast and more orography than in the SH. The NH thus exhibits stronger large scale wave activity and polar vortex and stratospheric variability (see Butchart et al. (2014) and Kidston et al. (2015) and references therein).*'

L273-287: This paragraph is very confusing and should be re-written to aid the readers understanding of the QBOs influence on the transport of ozone, and how this differs at different latitudes. This is a key point of the paper, and so spending some time clarifying this section will significantly improve the manuscript.
We have made amendments to this paragraph to improve understanding.
"We reiterate that the separation of positive and negative anomalies into those related to Easterly or Westerly QBO phases is clearest for the SH (Fig. 4a) and the corresponding, op posing, equatorial changes (Fig. 4b). The anti-correlated behaviour of anomalies between midlatitude and equatorial regions is consistent with previous studies investigating the relationship between the QBO and mid-latitude ozone variability (Zerefos et al., 1992; Randel et al., 1999; Strahan et al., 2015). We summarise the dynamical concept, in the context of these results, in the following (see Baldwin et al. (2001) and Choi et al. (2002) for detailed discussion). The QBO consists of downward propagating equatorial zonal winds; in the lower stratosphere this consists of a Westerly above an Easterly, or vice versa.  For Westerly above Easterly (i.e. the 15 hPa QBO is Westerly as identified by blue lines in Fig. 4) leads to a shear that induces a anomalously downward motion of air, and adiabatic warming (Fig. 1 of Choi et al. (2002)) and also to an anomalous increase in ozone; for an Easterly above a westerly, this leads to anomalously rising air and adiabatic cooling together with an associated ozone decreases; an equator-to-mid-latitude circulation forms to conserve mass (Randel et al., 1999; Polvani et al., 2010, Tweedy et al., 2017). At sub-tropical and mid-latitudes, the return of this meridional circulation draws ozone-rich air from above down into ozone poor regions, anomalously enhancing ozone (yellow, Fig. 4a,c). When Easterlies lie over Westerlies (blue, Fig. 4), the opposite circulation is set up, and ozone anomalies reverse."

Figure 4: Why does the red dashed line in the lower panels not have the same value where it intercepts the left and right hand side axes? The figure caption says the red dashed line is for January to October 2017, so the July value should be the same on both, as both are July 2017.
This is because each 12 month period is treated independently and normalised to September in each 12 month sequence. If there is a trend, or multi-year variability, or short-term fluctuations, it will not be the same level each yeah due to the normalisation in September. Of course, in reality they are connected and the same value in an absolute sense.

L311: Remove 'other'
Instead replaced with 'timescale'.

L325-334: The sensitivity of total column ozone trends to the choice of end year is also discussed by Weber et al. (2018) and Keeble et al. (2018), both of which could be cited here. Weber et al. has also now been published and the citation in the reference list and throughout the manuscript should be updated to reflect this.

We have included references, as well as Frith et al. (2014), to these sensitivity studies in the footnote that accompanied the point regarding the curve being 'locked-in' over certain timescales; we include Keeble et al., 2018, but note they did not demonstrate, though did discuss, the sensitivity of MLR to the data. However, we note that MLR sensitivity analyses are not directly comparable with the DLM technique in the sense that the whole period for which a linear trend is estimated in MLR is affected, whereas a limited timescale is affected in DLM due to a local change in the gradient of the non-linear curve close to the perturbation.

L373-284: The authors should clarify that here they are discussing ozone trends and not ozone values, which should not be affect by the addition of subsequent years.
Done; see additions to revised manuscript.

L387: change 'tropical' to 'tropics'
Done.

L401-403: Care must be taken here not to equate the success of the Montreal Protocol with increases in ozone. The Montreal Protocol is working – look at decreases to anthropogenic Cl – but it is not possible to say, without exploring the drivers of the changes, that increasing ozone in the upper stratosphere is evidence of that success. See also major comment above for use of ozone recovery and ozone increases.
An important but subtle point raised by the reviewer, which we agree with. We have preceded this point with (bold/italic):
*'An upper stratospheric increase is the expected result from long-term stratospheric chlorine reductions, a direct consequence of the Montreal Protocol and its amendments, though we do not explicitly attribute the cause of the increase to that here (for more on attribution see, for example WMO (2018). Indeed, the Montreal Protocol and its amendments will have been effective in reducing ozone losses throughout the atmosphere through reductions in CFC emissions, HCFCs and other ODSs. The lack of a positive trend since 1998 in the lower stratosphere, as opposed to the one clear in the upper stratosphere, is likely the consequence of other factors such as dynamical changes (Wargan et al., 2018).* These results once again reinforce  the conclusion that only the SH is affected by the 2017 *ozone* increase (lower stratosphere), that the Montreal Protocol  *appears* to be working (upper stratosphere), and that the decreases in the lower stratosphere at tropical and NH latitudes remain in place, but are not yet fully understood.'

L412-414: Consider rewording this sentence for clarity – as it is currently written, it is very hard to follow what is meant.
Rewritten:
'*The aim of this work is to assess the current state of, and trends in, stratospheric ozone. Improved knowledge of such trends, and the relevant forcing mechanisms and associated variability, will help to better constrain CCM projections of ozone to the end of the 21st Century.*'

L423-425: Clarify here that this is a result from a modelling study, not an extension of the observed record of ozone changes.
We have done this: 'A recent study used a CTM to reconstruct  the ozone timeseries *beyond the observational record available at the time to 2017* and found lower stratospheric ozone had rapidly increased in 2017 back to 1998 levels.'

L454: Replace 'spread on' with 'spread in'

Done.

L464: Replace 'predictions' with 'projections'
Done.

L465-484: There is an important feedback here, as ozone changes also affect the dynamics. Several CCM studies have highlighted the impacts of ozone depletion on stratospheric circulation, and as ozone starts to recover due to reductions of ODS, stratospheric dynamics will respond.
A good suggestion to add. We have now included the following after mention of a change in large scale stratospheric circulation: '*
[revised manuscript text omitted]

415  **ing end year**, although **the uncertainty** always u̶n̶c̶e̶r̶t̶a̶i̶n̶ **remains large** (Fig. 6); at mid-latitudes uncertainties **in the change since 1998** are smaller, but there has been a general shift towards more positive and significant increases, which is more-clearly reflected in the SH and quasi-global estimates. The evolution of the lower and whole stratospheric non-linear **ozone** trends mimic each other: south of 30°S, the end points of the negative changes have quickly increased in 2017 and

420  2018 **relative to 1998**, though remain negative in the lower stratosphere; at latitudes north of 30°S, the addition of 2017 and 2018 h̶a̶v̶e̶ **has** made little difference to the monotonic **ozone** decline; for 50–60°N, while flat, the additional years make little difference. The quasi-global lower stratospher̶e̶ic **ozone** continues to exhibit a monotonic decline that is still highly confident with 99% probability (Fig. 7 and Table 1), and **ozone abundances integrated over** the whole stratosphere **continues to**

425  remain lower **in 2018** than in 1998, though this is now with a probability of 86%; these trends are dominated by the tropical contribution (58%, latitude weighted) to the quasi-global change, whereas **the** NH and SH contribute 21% each. Even so, the NH changes do not appear affected by the recent large seasonally-dependent QBO variability.

     Figure 5 also confirms that the **gradients of the** non-linear curves are only affected by unmodelled

430  variance in years close to the end points, **typically within the last five years of the partial column timeseries considered here; the shape of the DLM curves** c̶h̶a̶n̶g̶e̶s̶ prior to the **final** l̶a̶s̶t̶ five years **of the DLM curves** are largely unaffected i̶n̶ ̶t̶h̶e̶ ̶p̶a̶r̶t̶i̶a̶l̶ ̶c̶o̶l̶u̶m̶n̶s̶. 
[revised manuscript text omitted]